# ATG7/GAPLINC/IRF3 axis plays a critical role in regulating pathogenesis of influenza A virus

Biao Chen[1,2☯], Guijie Guo[1,3☯], Guoqing Wang[1,3☯], Qianwen Zhu[1], Lulu Wang[1], Wenhao Shi[1], Song Wang[1,3], Yuhai Chen[2], Xiaojuan Chi[1], Faxin Wen[1], Mohamed Maarouf[2], Shile Huang[4], Zhou Yang[1], Ji-Long Chen[1,2,3]*

1 Key Laboratory of Animal Pathogen Infection and Immunology of Fujian Province, College of Animal Sciences, Fujian Agriculture and Forestry University, Fuzhou, People's Republic of China, 2 CAS Key Laboratory of Pathogenic Microbiology and Immunology, Institute of Microbiology, Chinese Academy of Sciences (CAS), Beijing, People's Republic of China, 3 Key Laboratory of Fujian-Taiwan Animal Pathogen Biology, College of Animal Sciences, Fujian Agriculture and Forestry University, Fuzhou, People's Republic of China, 4 Department of Biochemistry and Molecular Biology, Louisiana State University Health Sciences Center, Shreveport, Louisiana, United States of America

☯ These authors contributed equally to this work.
* chenjl@im.ac.cn, chenjilong@fafu.edu.cn

**Data Availability Statement:** All relevant data are within the Manuscript, Figures and Supporting Information files. The RNA-Seq data have been deposited in GEO public database under the accession number GSE211357.

## Abstract

Autophagy-related protein 7 (ATG7) is an essential autophagy effector enzyme. Although it is well known that autophagy plays crucial roles in the infections with various viruses including influenza A virus (IAV), function and underlying mechanism of ATG7 in infection and pathogenesis of IAV remain poorly understood. Here, *in vitro* studies showed that ATG7 had profound effects on replication of IAV. Depletion of ATG7 markedly attenuated the replication of IAV, whereas overexpression of ATG7 facilitated the viral replication. ATG7 conditional knockout mice were further employed and exhibited significantly resistant to viral infections, as evidenced by a lower degree of tissue injury, slower body weight loss, and better survival, than the wild type animals challenged with either IAV (RNA virus) or pseudorabies virus (DNA virus). Interestingly, we found that ATG7 promoted the replication of IAV in autophagy-dependent and -independent manners, as inhibition of autophagy failed to completely block the upregulation of IAV replication by ATG7. To determine the autophagy-independent mechanism, transcriptome analysis was utilized and demonstrated that ATG7 restrained the production of interferons (IFNs). Loss of ATG7 obviously enhanced the expression of type I and III IFNs in ATG7-depleted cells and mice, whereas overexpression of ATG7 impaired the interferon response to IAV infection. Consistently, our experiments demonstrated that ATG7 significantly suppressed IRF3 activation during the IAV infection. Furthermore, we identified long noncoding RNA (lncRNA) GAPLINC as a critical regulator involved in the promotion of IAV replication by ATG7. Importantly, both inactivation of IRF3 and inhibition of IFN response caused by ATG7 were mediated through control over GAPLINC expression, suggesting that GAPLINC contributes to the suppression of antiviral immunity by ATG7. Together, these results uncover an autophagy-independent mechanism by which ATG7 suppresses host innate immunity and establish a critical role for ATG7/GAPLINC/IRF3 axis in regulating IAV infection and pathogenesis.

**Funding:** This work was supported by National Natural Science Foundation of China (32030110 to JLC, U23A20235 to JLC) and National Key Research and Development Program of China (2021YFD1800205 to JLC).The funders had no role in study design, data collection and analysis, decision to publish, or preparation of the manuscript.

## Author summary

Influenza A virus (IAV) causes acute respiratory diseases in human and animals, posing a great threat to public health. Autophagy plays crucial roles in viral infections including IAV, but mechanisms underlying interaction between autophagy and IAV remain ambiguous. Particularly, function and underlying mechanisms of ATG7, an essential autophagy effector enzyme, in viral infections are largely unexplored, and little information is available about relationship between ATG7 and IAV pathogenesis. Here, we used *in vitro* and *in vivo* models to address ATG7 function in the IAV infection and pathogenesis. We found that forced expression of ATG7 facilitates the viral replication, while depletion of ATG7 attenuates viral replication and renders mice more resistant to IAV or pseudorabies virus (PRV) infection. Importantly, we identify that ATG7 suppresses IRF3 activation and interferon production via lncRNA GAPLINC, revealing an autophagy-independent mechanism whereby ATG7 restrains host innate immunity and unveiling a critical role of ATG7/GAPLINC/IRF3 axis in regulating IAV pathogenesis. Moreover, our observations suggest that ATG7 may positively regulate the expression of GAPLINC via suppression of NF-κB activation during IAV infection. Together, these results reveal that ATG7 has multiple biological roles beyond autophagy, and provide an important insight into the complicated interplay between host and IAV.

## Introduction

Autophagy is an evolutionarily-conserved cellular degradative pathway, which mediates the degradation of encapsulated cytoplasmic material via the endolysosomal system [1,2]. Autophagy is involved in a variety of biological processes such as the maintenance of cellular homeostasis, cell differentiation, and host defense against invading pathogens [3–5]. Increasing evidence suggests that autophagy plays important roles in viral pathogenesis, although the effects of autophagy on viral replication and the outcome of viral infection differ depending on the viruses and the host cells [6,7]. On the one hand, hosts could utilize their own autophagy to prevent viral infection and pathogenesis [8–10]. For instance, IFN-β-induced endoplasmic reticulum protein SCOTIN interacts with hepatitis C virus (HCV) non-structural protein 5A (NS5A), and targets NS5A to autophagosomes for degradation, hence restricting HCV replication [11]. In addition, autophagy has been reported to prevent viral invasion through activating innate immune responses [12,13]. A recent study has found that TRIM14 could recruit the deubiquitinase USP14 to remove K48-linked ubiquitin chains of cyclic GMP-AMP synthase (cGAS), an essential DNA virus sensor that triggers type I interferon (IFN) signaling, which leads to the inhibition of p62-mediated autophagic degradation of cGAS, therefore promoting type I IFN response [14]. Besides, autophagy is also involved in coordinating adaptive immunity by promoting antigen presentation, which is essential for the elimination of invading viruses. Deletion of the key autophagy gene ATG5 in dendritic cells (DCs) of mice results in a significant impairment of CD4$^+$ T cell priming after herpes simplex virus (HSV) infection, thereby accelerating the pathogenesis of HSV in the animal [15]. These investigations suggest that autophagy may function through different mechanisms to regulate viral infection and pathogenesis.

On the other hand, emerging data suggest that some viruses have evolved several strategies to hijack and manipulate host autophagy for their own survival and proliferation in host cells [16]. For instance, severe acute respiratory syndrome coronavirus-2 (SARS-CoV-2) viral

protein ORF3a interacts with VPS39, and impedes the association of the homotypic fusion and protein sorting (HOPS) complex with STX17 or RAB7, thereby blocking the fusion of autophagosomes with lysosomes and eventually inhibiting the autophagy activity [17,18]. This may be a mechanism employed by SARS-CoV-2 to escape host lysosome degradation. In addition, autophagy has been reported to enhance the replication of some viruses such as coxsackievirus B3 (CVB3), HCV, and poliovirus, as genetic or pharmacological inhibition of autophagy decreases viral yields [19–21], implying that some viruses can take advantage of host autophagy for their infection and replication.

Influenza A virus (IAV) is an important member of the Orthomyxoviridae family, which causes acute respiratory diseases in human and a lot of animals, posing a great threat to the health of humans and animals. It has been shown that the PB1 protein of IAV is associated with the selective autophagic receptor neighbor of BRCA1 (NBR1), and the latter recognizes ubiquitinated MAVS and targets it for autophagic degradation, consequently restraining the RIG-I-MAVS-mediated innate immune signaling and facilitating viral infection [22]. Similarly, the NP protein of IAV is recently reported as a critical regulator of mitophagy, and NP-mediated mitophagy leads to the degradation of MAVS, thereby blocking MAVS-mediated antiviral signaling and promoting viral replication [23].

Notably, it has been shown that autophagy-related proteins (ATG) have extensive biological importance beyond autophagic elimination [24]. For instance, the ATG5-ATG12/ATG16L1 complex and ATG7, but not the degradative activity of autophagy, are required for the antiviral activity of IFN-γ against murine norovirus (MNV) infection in macrophages through involvement in IFN-γ-mediated inhibition of MNV replication complex formation [25]. Additionally, ATG9a could co-localize with stimulator of IFN genes (STING), an essential signal transducer required for dsDNA-triggered innate immune responses [26]. Depletion of ATG9a enhances the assembly of STING and TANK-binding kinase 1 (TBK1) after dsDNA stimulation, leading to aberrant activation of innate immune responses [26], suggesting a role of ATG9a in the regulation of innate signaling which is independent from its implication in autophagy. Besides, ATG7, an essential autophagy effector enzyme, interacts with p53 and inhibits the expression of pro-apoptotic factors such as Noxa, Puma and Bax, which is independent on its E1-like enzymatic activity [27]. Accordingly, ATG7-null mouse embryonic fibroblasts displayed augmented DNA damage [27]. In another study, ATG7 has been shown to directly interact with caspase-9 and suppress the pro-apoptotic activity of caspase-9, which is not related to its function in the autophagic process [28]. Importantly, a direct association between ATG7 dysfunction and disease was recently established in patients with biallelic ATG7 variants [29]. However, the functional repertoire and underlying mechanisms of ATG7 in viral infection and pathogenesis are largely unexplored. Particularly, little information is available about relationship between ATG7 and IAV pathogenesis.

In this study, we used *in vitro* and *in vivo* models to address ATG7 function in the viral pathogenesis. The results show that forced expression of ATG7 facilitates the viral replication, while depletion of ATG7 attenuates viral replication and renders mice more resistant to IAV or pseudorabies virus (PRV) infection. Mechanistically, ATG7 promotes the viral replication in autophagy-dependent and -independent manners. Importantly, we identify that ATG7 enhances the expression of lncRNA GAPLINC, resulting in the suppression of IRF3 activation and IFN production, thereby restraining host antiviral immunity. These findings reveal that ATG7 regulates viral infection and pathogenesis via multiple mechanisms, including an autophagy-independent mechanism involving the ATG7/GAPLINC/IRF3 axis.

## Results

### Altering ATG7 expression has profound effects on replication of IAV

Numerous studies have shown that autophagy plays important roles in viral infections. ATG7 is an essential autophagy effector enzyme, but its functional involvement in the IAV pathogenesis is largely unknown. To define the role of ATG7 in IAV infections, we generated A549 cells stably expressing specific shRNAs targeting ATG7 (sh1-ATG7 and sh2-ATG7) or luciferase control (sh-Luc) using lentiviral vectors. The protein levels of ATG7 were dramatically reduced in A549 cells expressing ATG7 shRNAs compared with that in cells expressing control shRNA (**S1A Fig**). Then, ATG7 knockdown and control cells were infected with influenza virus A/WSN/33 (H1N1), and viral loads were determined by hemagglutination (HA) and plaque forming assays. Notably, ATG7 knockdown significantly decreased IAV titers in the cells compared to the control (**Fig 1A and 1B**). In line with this, the levels of viral nucleoprotein (NP) protein were dramatically reduced in ATG7 knockdown cells (**Fig 1C**). These data indicated that depletion of ATG7 impeded the replication of IAV in A549 cells. Furthermore, we infected ATG7 knockdown and control A549 cells with other strains of IAV including A/PR8/34 (H1N1) and H9N2 subtype virus, and similar results were observed in these experiments (**Fig 1D–1G**). Moreover, Sendai virus (SeV) was also employed, and consistently, ATG7 knockdown hindered the replication of the virus (**S1B and S1C Fig**).

On the other hand, we evaluated the effect of ATG7 overexpression on the viral replication. A549 cell lines stably expressing ATG7 or empty vector (EV) were generated by using lentiviral vectors. These cells were then infected with influenza virus (H1N1 and H9N2) or SeV, and replication of the viruses was examined. As expected, ATG7 overexpression markedly increased IAV titers and elevated viral NP mRNA levels in A549 cells compared to control, in response to infection with H1N1 IAVs including WSN (**Figs 1H, 1I and S1D**) and PR8 (**Figs 1J, 1K and S1E**). Similarly, overexpression of ATG7 also resulted in enhanced H9N2 IAV replication (**Fig 1L**), and a significant increase in SeV NP RNA levels in cells after the viral infection (**S1F and S1G Fig**). Together, these observations indicate that altering ATG7 expression has profound effects on the replication of IAV, as well as SeV.

### *In vivo* deficiency of ATG7 significantly impairs virulence of IAV in mice

To further substantiate the function of ATG7 in IAV infection and pathogenesis under a more elaborate and physiological circumstance, we employed ATG7 conditional knockout (CKO) mice, as the germ line knockout of ATG7 is died within 1 day after birth [30]. ATG7$^{flox/flox}$ mice were crossed with transgenic UBC-CreERT2 mice. ATG7$^{flox/flox}$/UBC-CreERT2 mice were treated with tamoxifen to induce the Cre recombinase that mediates the knockout of ATG7. We observed that ATG7 protein levels were profoundly decreased in the liver, spleen, lung, kidney, and thymus of ATG7$^{flox/flox}$/UBC-CreERT2 mice treated with tamoxifen (**Fig 2A**). The mice were then infected with influenza virus A/PR8/34 (H1N1), and the viral replication was examined. ATG7 CKO mice displayed slower body weight loss and better survival than control mice during IAV infection (**Fig 2B and 2C**). In addition, ATG7 CKO mice exhibited a lower degree of lung injury caused by IAV infection than control mice (**Fig 2D and 2E**), suggesting that deficiency of ATG7 attenuated IAV replication in mice. Consistent with these observations, IAV titers were significantly decreased and the viral NP protein levels were clearly reduced in lung tissues from ATG7 CKO mice compared with those in the tissues of control mice (**Fig 2F and 2G**). These results reveal that depletion of ATG7 renders mice more resistant to the IAV infection.

Since mouse is an ideal animal model for infection with PRV, a DNA virus, we also determined whether ATG7 was involved in the pathogenesis of PRV. To this end, control and

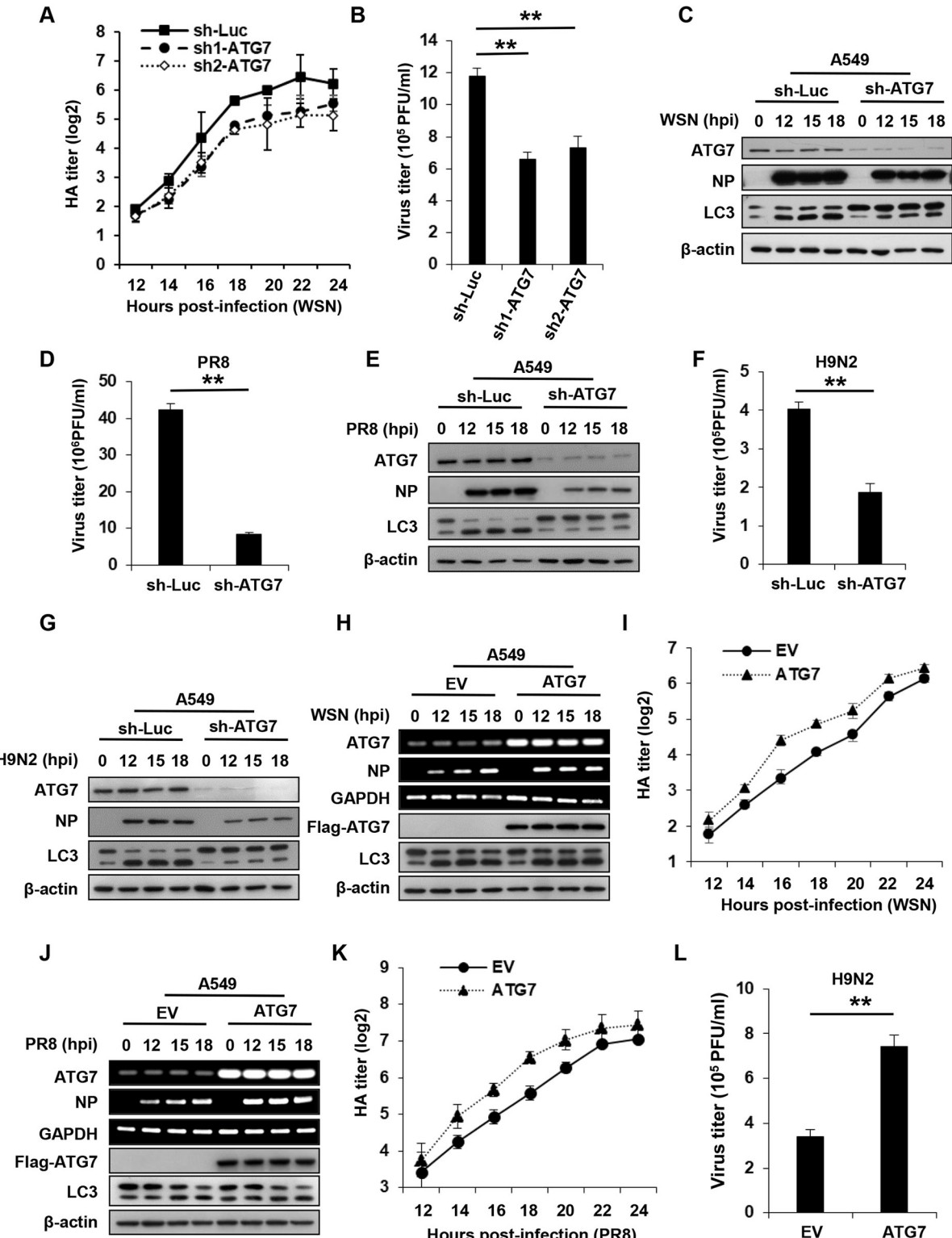

**Fig 1. Altering ATG7 expression has profound effects on the replication of IAV. (A and B)** Control and ATG7 knockdown A549 cells were infected with influenza virus A/WSN/33 (MOI = 0.5), and the supernatants were collected for hemagglutination (HA) assay (A) and plaque forming assay (B). **(C)** Control and ATG7 knockdown A549 cells were infected with influenza virus A/WSN/33 (MOI = 0.5) for the indicated time, and viral NP protein levels in the cells were examined by Western blotting. **(D and E)** Control and ATG7 knockdown A549 cells were infected with PR8 (MOI = 0.2), and the supernatants were collected at 16 hpi for plaque forming assay (D). Western blotting was

performed to detect viral NP protein levels in the cells (E). **(F and G)** Control and ATG7 knockdown A549 cells were infected with H9N2 influenza virus (MOI = 1). The supernatants were collected at 16 hpi for plaque forming assay (F), and Western blotting was performed to detect viral NP protein levels in the cells (G). **(H and I)** Control and ATG7 overexpressing A549 cells were infected with WSN (MOI = 0.5) for the indicated time. Viral NP RNA levels in the cells were examined by RT-PCR (H), and the supernatants were collected for HA assay (I). **(J and K)** Control and ATG7 overexpressing A549 cells were infected with PR8 (MOI = 0.2) for the indicated time. RT-PCR was performed to test viral NP RNA levels in the cells (J), and the supernatants were collected for HA assay (K). **(L)** Control and ATG7 overexpressing A549 cells were infected with H9N2 influenza virus (MOI = 1) for 16 h. The supernatants were collected for plaque forming assay. RT-PCR and Western blotting data were repeated independently three times with similar results. Shown are representative data of three biologically independent experiments. Data are presented as means ± SD from three independent experiments, $^{**}p < 0.01$.

ATG7 CKO mice were infected with PRV, and the effect of ATG7 deficiency on PRV replication was evaluated. A time course study showed that ATG7 CKO mice had slower body weight loss and an increased survival rate than control mice challenged with PRV (**Fig 2H and 2I**). Accordingly, lower viral gE RNA and protein levels were detected in the lung, brain, and liver derived from ATG7 CKO mice than those in the tissues of control mice after PRV infection (**Figs 2J, 2K and S2A–S2D**). Together, the results suggest that ATG7 facilitates pathogenesis of both IAV and PRV in mice, implying that ATG7 might function as a pro-viral host factor for *in vivo* pathogenesis of a broad spectrum of viruses.

## ATG7 promotes the replication of IAV in autophagy-dependent and -independent manners

Next, we determined how ATG7 was implicated in IAV infection and pathogenesis. Since ATG7 is an essential component of autophagy machinery, in concert with other ATG proteins which regulate autophagy process, we asked whether ATG7 affected viral replication through the autophagy pathway. To address this, control and ATG7 knockdown A549 cells were treated with hydroxychloroquine sulfate (HCQ), a potent inhibitor of autophagy that prevented lysosomal acidification and thereby interfering with the autophagic process [31]. The cells were then infected with influenza virus A/PR8/34, and viral replication was examined by plaque forming assay. We found that treatment with HCQ did not completely block the inhibitory effect of ATG7 deficiency on IAV replication, but still displayed the ATG7 knockdown-induced inhibition of IAV replication (**Figs 3A and S3A**). These data indicated that inhibition of the autophagy by HCQ could not deter the impairment of IAV replication caused by ATG7 depletion, implying that ATG7 might regulate the replication of IAV, only to some extent, requiring intact autophagy. To confirm this, we treated control and ATG7 knockdown cells with another inhibitor of autophagy, MRT68921 that specifically disrupts autophagosome maturation and blocks autophagic flux [32], followed by IAV infection. Similarly, MRT68921 treatment failed to eliminate the ATG7 knockdown-induced inhibition of IAV replication in the cells (**Figs 3B and S3B**). Additionally, silencing of BECN1 or ATG3, two key proteins involved in autophagy [33,34], could not block the attenuation of IAV replication caused by loss of ATG7 in the knockdown cells (**S3C and S3D Fig**). Moreover, we observed that overexpression of ATG7 still promoted the replication of IAV compared with control cells, although BECN1 or ATG3 was depleted in the cells (**S3E and S3F Fig**). On the other hand, overexpression of ATG7 attenuated the inhibitory effect of HCQ and MRT68921 on IAV replication in A549 cells (**Fig 3C and 3D**). The results suggest that ATG7 may regulate IAV replication through autophagy-dependent and -independent mechanisms.

To further test the autophagy-dependent and -independent mechanisms underlying regulation of IAV replication by ATG7, we constructed two ATG7 mutants devoid of the ability to efficiently regulate autophagy, including the active-site mutant (ATG7$^{C572S}$) [35], and the mutant defective in the formation of the E2-substrate intermediate of ATG3 and LC3

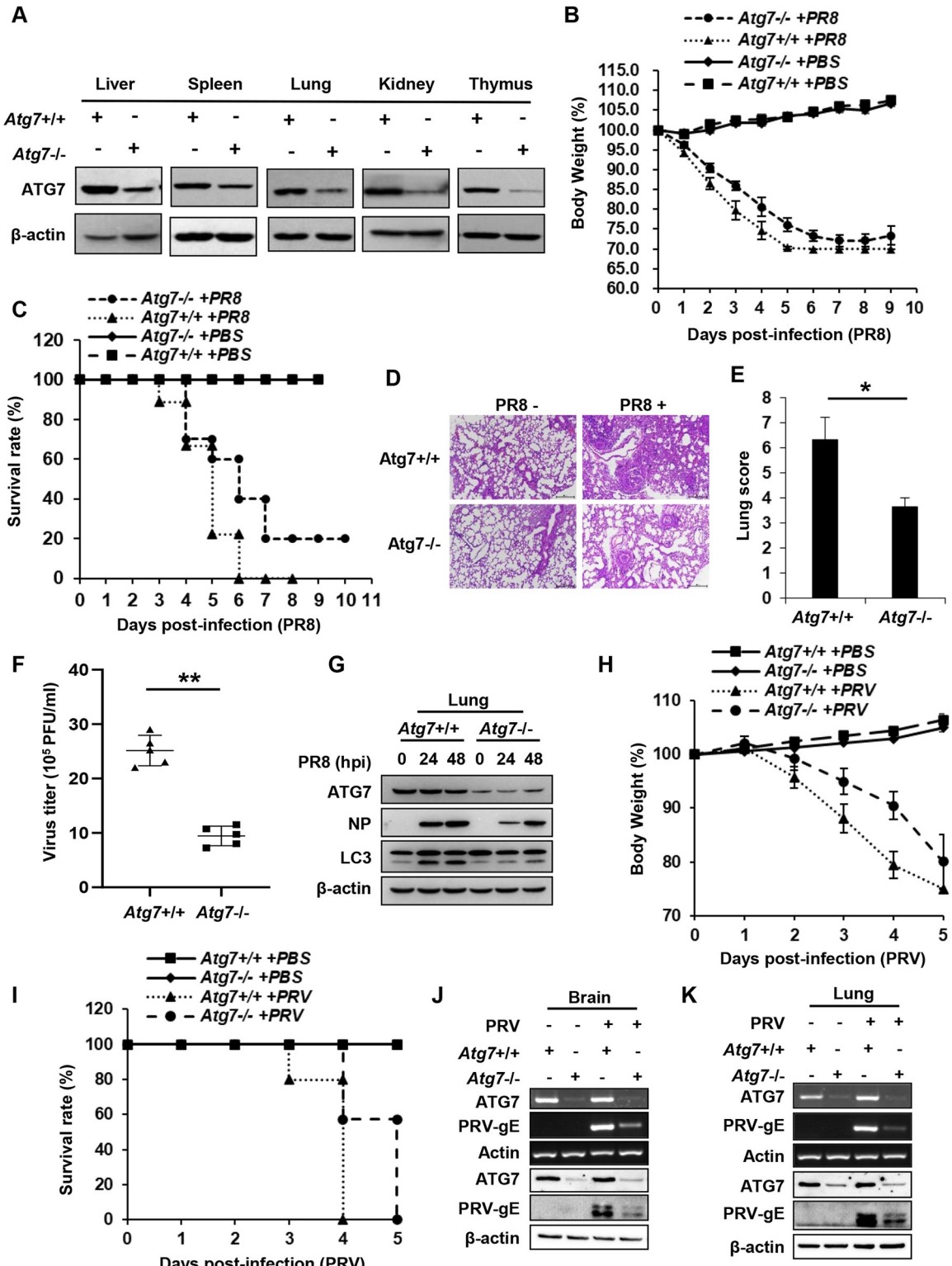

**Fig 2. In vivo deficiency of ATG7 significantly impairs virulence of IAV and PRV in mice. (A)** ATG7 protein levels in indicated tissues of WT and ATG7 conditional knockout (CKO) mice were examined by Western blotting. **(B-G)** WT and ATG7 CKO mice were injected intranasally with $5×10^4$ plaque-forming units (PFU) of PR8. The body weight loss (B), and the survival rate (C) of mice were monitored. The histopathologic changes in the lungs from WT and ATG7 CKO mice at 2 dpi were determined by HE staining (D), and scored for disease severity (E). Scale bar, 200 μm. The viral titers and NP protein levels in the lungs from WT and ATG7 CKO mice were

determined by plaque forming assay (F) and Western blotting (G) respectively. (**H-K**) WT and ATG7 CKO mice were injected intramuscularly with $1\times10^6$ PFU of PRV. The body weight loss (H), and survival rate (I) of mice were monitored. Viral gE mRNA and protein levels in the brains from WT and ATG7 CKO mice at 2 dpi were determined by RT-PCR and Western blotting respectively (J). Viral gE mRNA and protein levels in the lungs of mice at 2 dpi were detected by RT-PCR and Western blotting respectively (K). RT-PCR and Western blotting data were repeated independently three times with similar results. Shown are representative data of three biologically independent experiments. Data are presented as means ± SD from three independent experiments, $^{**}p < 0.01$.

(ATG7$^{FAPtoDDD}$) [36]. Then, we generated A549 cells expressing either wild type (WT) ATG7, each ATG7 mutant, or EV control, followed by infection with IAV (**Fig 3E**). As predicted, overexpression of ATG7 WT led to an enhanced replication of IAV compared to EV control (**Fig 3F**). Interestingly, overexpression of ATG7 mutants also significantly increased IAV replication (**Fig 3F**), although the autophagy was suppressed in cells expressing the ATG7 mutants (**Fig 3E**), implying an autophagy-independent pathway utilized by ATG7 to facilitate the IAV replication. However, we noticed that the stimulatory effect of ATG7 WT on IAV replication was stronger than the ATG7 mutants (**Fig 3F**). Furthermore, we generated ATG7 knockdown A549 cells re-expressing either WT or mutants of ATG7, followed by infection with IAV (**Fig 3G**). Indeed, ATG7 knockdown significantly decreased IAV titers (**Fig 3H**). Notably, add-back of either WT or mutants of ATG7 recovered the attenuated IAV replication caused by loss of ATG7 in ATG7 knockdown cells, despite defected autophagy (**Fig 3H**). Collectively, these results support that ATG7 promotes the replication of IAV in both autophagy-dependent and -independent manners.

## ATG7 inhibits IAV-induced expression of IFNs *in vitro* and *in vivo*

Next, we sought to dissect the molecular mechanisms by which ATG7 promotes the viral replication. For this, we performed RNA sequencing (RNA-Seq) to analyze differentially expressed mRNAs between control and ATG7 knockdown A549 cells infected with influenza virus A/PR8/34 (H1N1). Notably, our RNA-Seq analysis revealed that the expression levels of type I and III IFNs, which play central roles in restricting viral infections, were significantly increased in ATG7 knockdown A549 cells as compared to the control cells upon IAV infection (**Figs 4A and S4A**). This finding was further confirmed by analysis of real-time PCR (**Figs 4B–4D and S4B and S4C**). Consistently, it was shown that ATG7 knockdown significantly elevated protein levels of IFN-β in the cells infected with IAV (**Fig 4E**).

Next, we asked whether ATG7 regulated the IFN expression via autophagy-dependent or -independent mechanism induced by IAV. For this, we used poly(I:C), a dsRNA mimic to induce IFNs expression without induction of autophagy, to examine its influence on the levels of IFN-β in control and ATG7 knockdown A549 cells. Interestingly, disruption of ATG7 expression also led to a significant increase in the expression of IFN-β induced by poly(I:C) (**Fig 4F and 4G**). These results suggest that ATG7-mediated suppression of IFN response involves an autophagy-independent mechanism. Moreover, SeV was employed, and similarly, the virus-induced expression of IFN-β, IL-28 and IL-29 was obviously upregulated in the absence of ATG7 (**S4D–S4G Fig**). Additionally, we measured the levels of IFNs in ATG7 overexpressing A549 cells and the control challenged with IAV. As expected, overexpression of ATG7 resulted in a significant decrease in mRNA expression of IFN-β, IL-28 and IL-29 (**Figs 4H–4J and S4H**), and in protein levels of IFN-β upon IAV infection (**Fig 4K**).

For *in vivo* analysis, the levels of IFNs were examined in the lungs of ATG7 CKO or WT mice challenged with IAV infection. Consistent with the data obtained from *in vitro* experiments, *in vivo* studies showed that ATG7 CKO mice had a significantly increased production of IFN-β and IL-28 in the lungs compared with the control mice infected with IAV (**Figs 4L–4N**

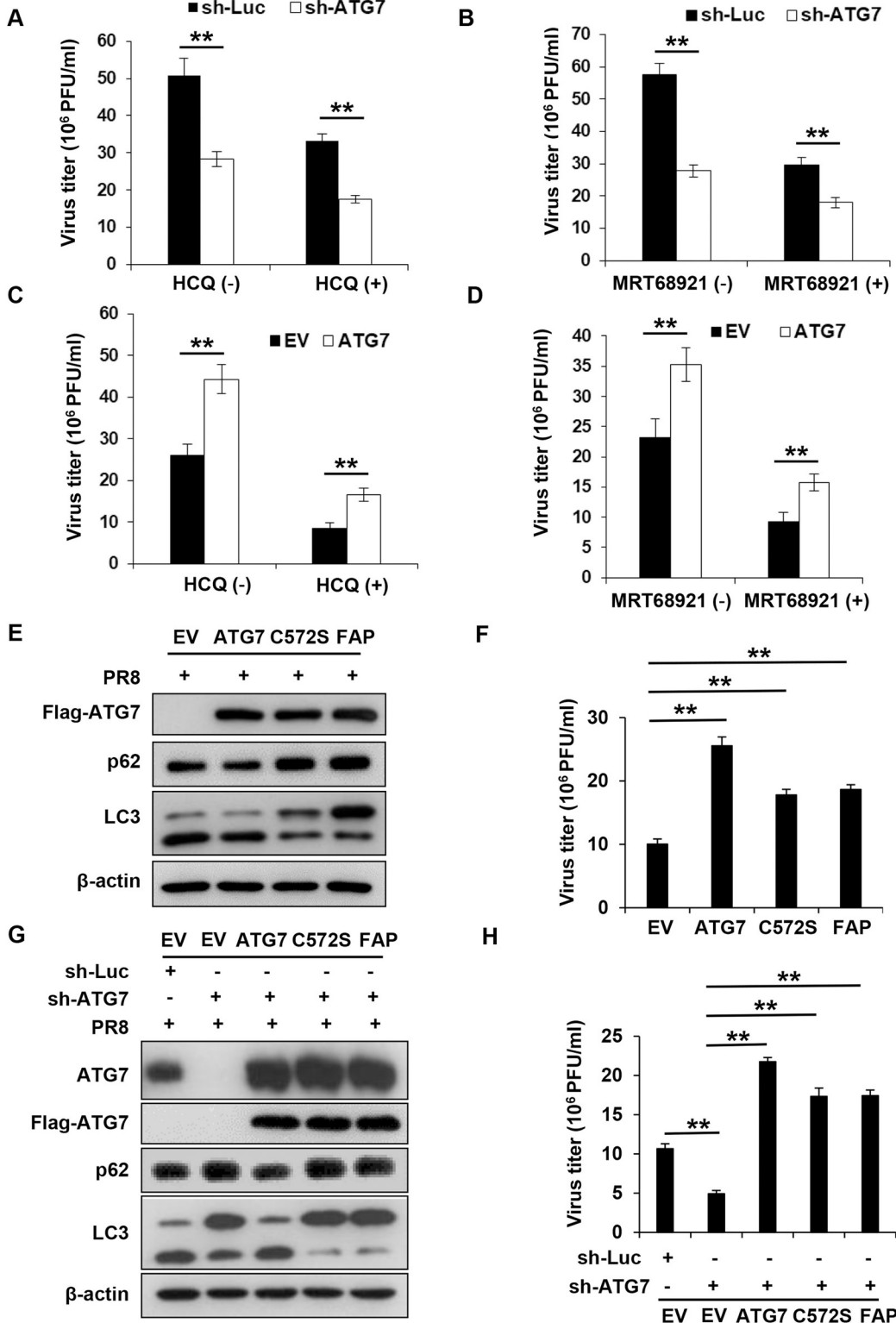

**Fig 3. ATG7 promotes the replication of IAV in autophagy-dependent and -independent manners. (A)** Control and ATG7 knockdown A549 cells were pretreated with DMSO or HCQ (20 μM) for 3 h, and then infected with PR8 (MOI = 0.2) for 16 h. The supernatants were collected for plaque forming assay. **(B)** Control and ATG7 knockdown A549 cells were pretreated with DMSO or MRT68921 (5 μM) for 3 h, and then infected with PR8 (MOI = 0.2) for 16 h. The supernatants were collected and subjected to plaque forming assay. **(C and D)** Control and ATG7 overexpressing A549

cells were pretreated with HCQ (20 μM) (C) or MRT68921 (5 μM) (D) for 3 h, and then infected with PR8 (MOI = 0.2) for 16 h. The supernatants were collected and subjected to plaque forming assay. **(E and F)** Control and A549 cells overexpressing WT, or mutants of ATG7 (ATG7$^{C572S}$, ATG7$^{FAPtoDDD}$) were infected with PR8 (MOI = 0.2) for 16 h (E). The supernatants were collected for plaque forming assay (F). **(G and H)** Control and ATG7 knockdown A549 cells re-expressing WT, or mutants of ATG7 (ATG7$^{C572S}$, ATG7$^{FAPtoDDD}$) were infected with PR8 (MOI = 0.2) for 16 h (G). The supernatants were collected for plaque forming assay (H). Western blotting data were repeated independently three times with similar results. Shown are representative data of three biologically independent experiments. Data are presented as means ± SD from three independent experiments, $^*p < 0.05$, $^{**}p < 0.01$.

and **S4I**). Together, both *in vitro* and *in vivo* data reveal that ATG7 inhibits the production of IFNs during the IAV infection.

## ATG7 suppresses IRF3 activation in an autophagy-independent manner during the IAV infection

Next, we further investigated the mechanisms by which ATG7 restrains the IFN response. Since RLR-dependent pathway is a main innate immune signaling activated by IAV infection to trigger the production of interferons, we explored whether ATG7 could suppress this signaling. To this end, IFN-β luciferase reporter system was employed, and ATG7 overexpressing cells and controls were transfected with the reporter and either RIG-I, MAVS, TBK1, WT IRF3 or its 5D mutant expression plasmid. The results showed that ATG7 significantly inhibited RIG-I-, MAVS-, TBK1-, and WT IRF3-mediated IFN-β luciferase activation, as ATG7 overexpressing cells exhibited lower IFN-β luciferase activity than control cells, whereas ATG7 had no significant effect on IFN-β luciferase activity induced by the 5D mutant, an active form of IRF3 (**Fig 5A**), implying that ATG7 restrains RLR signaling by suppressing IRF3 activation.

IRF3 is a key transcriptional factor implicated in the innate immune response to viral infections. This drove us to further evaluate effect of ATG7 on the activation of IRF3 upon viral infection. First, control and ATG7 knockdown A549 cells were transfected with poly(I:C), a dsRNA mimic to activate IRF3 without induction of autophagy. It was observed that ATG7 knockdown led to a significant increase in the phosphorylation levels of IRF3 induced by poly(I:C) (**Fig 5B**). Then, we examined the phosphorylation of IRF3 in control and ATG7 knockdown A549 cells infected with IAV. As shown in **Fig 5C**, higher levels of phosphorylated IRF3 were found in ATG7 knockdown cells than those in control cells after IAV infection. Consistently, the nuclear translocation of IRF3 was enhanced in ATG7 knockdown A549 cells compared with control cells following viral infection (**S5A Fig**). In contrast, knockdown of BECN1 or ATG3 had no significant effect on the phosphorylation of IRF3 compared with control cells after IAV infection (**S5B and S5C Fig**). These results revealed that depletion of ATG7 enhanced the activation of IRF3 during the viral infection. Moreover, we detected the phosphorylation status of IRF3 in control and ATG7 overexpressing A549 cells after poly(I:C) transfection or IAV infection. Indeed, overexpression of ATG7 resulted in a significant decrease in the phosphorylation of IRF3 induced by either poly(I:C) or IAV (**Fig 5D and 5E**). In addition, we generated ATG7 knockdown A549 cells re-expressing an shRNA-resistant form of ATG7, followed by transfection with poly(I:C) (**S5D and S5E Fig**). ATG7 knockdown led to a significant increase in the phosphorylation levels of IRF3 induced by poly(I:C), while re-introduction of ATG7 reversed the enhanced IRF3 phosphorylation caused by loss of ATG7 in the cells treated with poly(I:C) (**S5D and S5E Fig**). These observations indicate that ATG7 facilitates IAV replication likely through suppression of IRF3 activation and subsequent IFNs expression. Remarkably, the fact that ATG7 can impair the poly(I:C)-induced IRF3 phosphorylation suggests that suppression of IRF3 by ATG7 may be autophagy-independent.

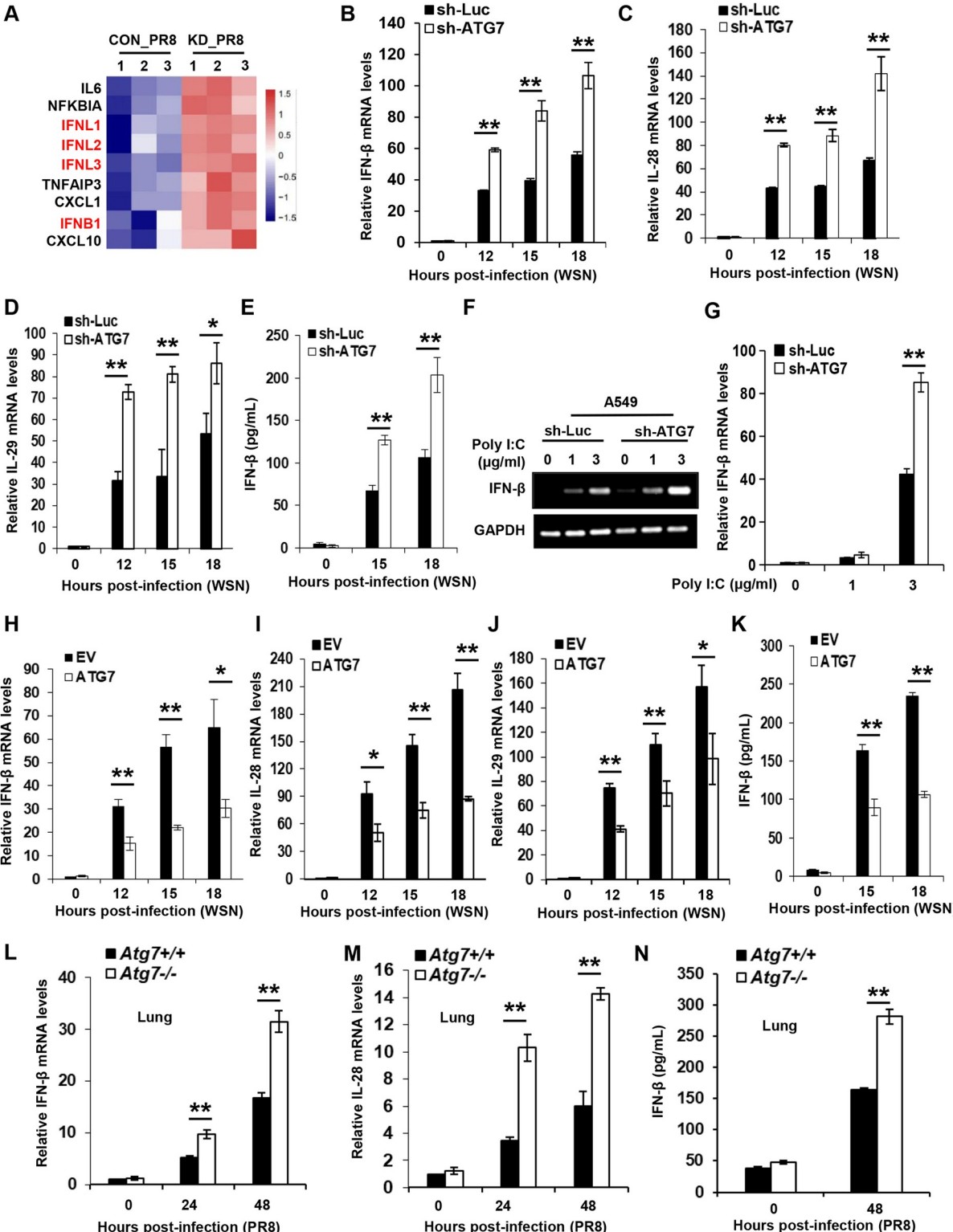

**Fig 4. ATG7 inhibits IAV-induced expression of IFNs in vitro and in vivo.** (**A**) Transcriptome RNA sequencing analysis of control and ATG7 knockdown A549 cells infected with PR8 (MOI = 0.5) for 16 h. (**B-E**) Control and ATG7 knockdown A549 cells were infected with WSN (MOI = 0.5) for 0, 12, 15, and 18 h. The RNA levels of IFN-β (B), IL-28 (C), or IL-29 (D) were examined by qRT-PCR. GAPDH was chosen as a reference gene for internal standardization. The IFN-β protein levels in the supernatants were examined by ELISA (E). (**F-G**) Control and ATG7 knockdown A549 cells were transfected with poly(I:C) for 4 h. The RNA levels of IFN-β were examined by RT-PCR (F)

and qRT-PCR (G) respectively. **(H-K)** Control and ATG7 overexpressing A549 cells were infected with WSN (MOI = 0.5) for 0, 12, 15, and 18 h. The RNA levels of IFN-β (H), IL-28 (I), or IL-29 (J) were examined by qRT-PCR, and the IFN-β protein levels in the supernatants were examined by ELISA (K). **(L-N)** WT and ATG7 CKO mice were infected with $5\times10^4$ PFU of PR8 for 0, 24, and 48 h. The RNA levels of IFN-β (L), and IL-28 (M) in the lungs of mice were determined by qRT-PCR, and the IFN-β protein levels were examined by ELISA (N). Data are presented as means ± SD from three independent experiments, $*p < 0.05$, $**p < 0.01$.

Thus, we further asked whether ATG7 regulated the activation of IRF3 via modulating the autophagy process. For this, we employed two ATG7 mutants lacking the ability to regulate autophagy, including the active-site mutant (ATG7$^{C572S}$) [35], and the mutant defective in the formation of the E2-substrate intermediate of ATG3 and LC3 (ATG7$^{FAPtoDDD}$) [36]. We generated A549 cells expressing either WT ATG7, each ATG7 mutant, or EV control, followed by infection with IAV. Overexpression of WT ATG7 caused a diminished phosphorylation of IRF3 compared to EV control (**Fig 5F**). Of note, comparable phosphorylation levels of IRF3 were observed between cells overexpressing WT and ATG7 mutants, although the autophagy was dampened in cells expressing ATG7 mutants (**Fig 5F**). We also generated ATG7 knockdown A549 cells re-expressing either WT or mutants of ATG7 followed by IAV infection, and the activation of IRF3 was evaluated (**Fig 5G**). In response to IAV infection, silencing of ATG7 resulted in a significant increase of IRF3 phosphorylation compared with the control, while re-introduction of either WT or mutants of ATG7 reversed the enhanced IRF3 phosphorylation caused by loss of ATG7 in the knockdown cells (**Fig 5G**). Together, these data imply that there exists an autophagy-independent mechanism underlying ATG7-mediated inactivation of IRF3 during the IAV infection.

## Identification of lncRNA GAPLINC as a critical regulator involved in the promotion of IAV replication by ATG7

There is increasing evidence revealing that long noncoding RNAs (lncRNAs) play critical roles in regulating innate immunity and IAV pathogenesis [37–39]. To dissect the autophagy-independent mechanism by which ATG7 promotes the viral replication, we performed RNA-Seq to analyze differentially expressed lncRNAs between control and ATG7 knockdown A549 cells infected with influenza virus A/PR8/34 (H1N1) (**Figs 6A and S6A**). Strikingly, the expression of an lncRNA, gastric adenocarcinoma predictive long intergenic noncoding RNA (GAPLINC), which has recently been described as an important modulator of the immune response during endotoxic shock [40], decreased in ATG7 knockdown A549 cells compared with the controls upon IAV infection (**Fig 6A and 6B**). Importantly, ATG7 CKO mice also showed a significantly decreased expression of GAPLINC in the lungs compared with the control mice after the IAV infection (**Fig 6C**). In contrast, knockdown of BECN1 or ATG3 had no significant effect on the expression of GAPLINC compared with control cells upon IAV infection (**S6B and S6C Fig**). On the other hand, overexpression of ATG7 led to an enhanced expression of GAPLINC in cells (**Fig 6D**), suggesting a possibility that GAPLINC may contribute to the ATG7-mediated regulation of IAV replication.

Thus, we further examined the expression of GAPLINC during IAV infection. We observed that GAPLINC was downregulated in a virus dose- and infection time-dependent manner (**S6D and S6E Fig**). Since it was reported that GAPLINC expression was negatively regulated by NF-κB signaling [40], we determined whether downregulation of GAPLINC was caused by activated NF-κB during the IAV infection. Indeed, such GAPLINC downregulation was impaired in A549 cells treated with an NF-κB inhibitor BAY11-7082 as compared to non-treated cells upon IAV infection (**S6F Fig**). This finding prompted us to evaluate the role of NF-κB in regulation of GAPLINC expression by ATG7. Control and ATG7 knockdown A549

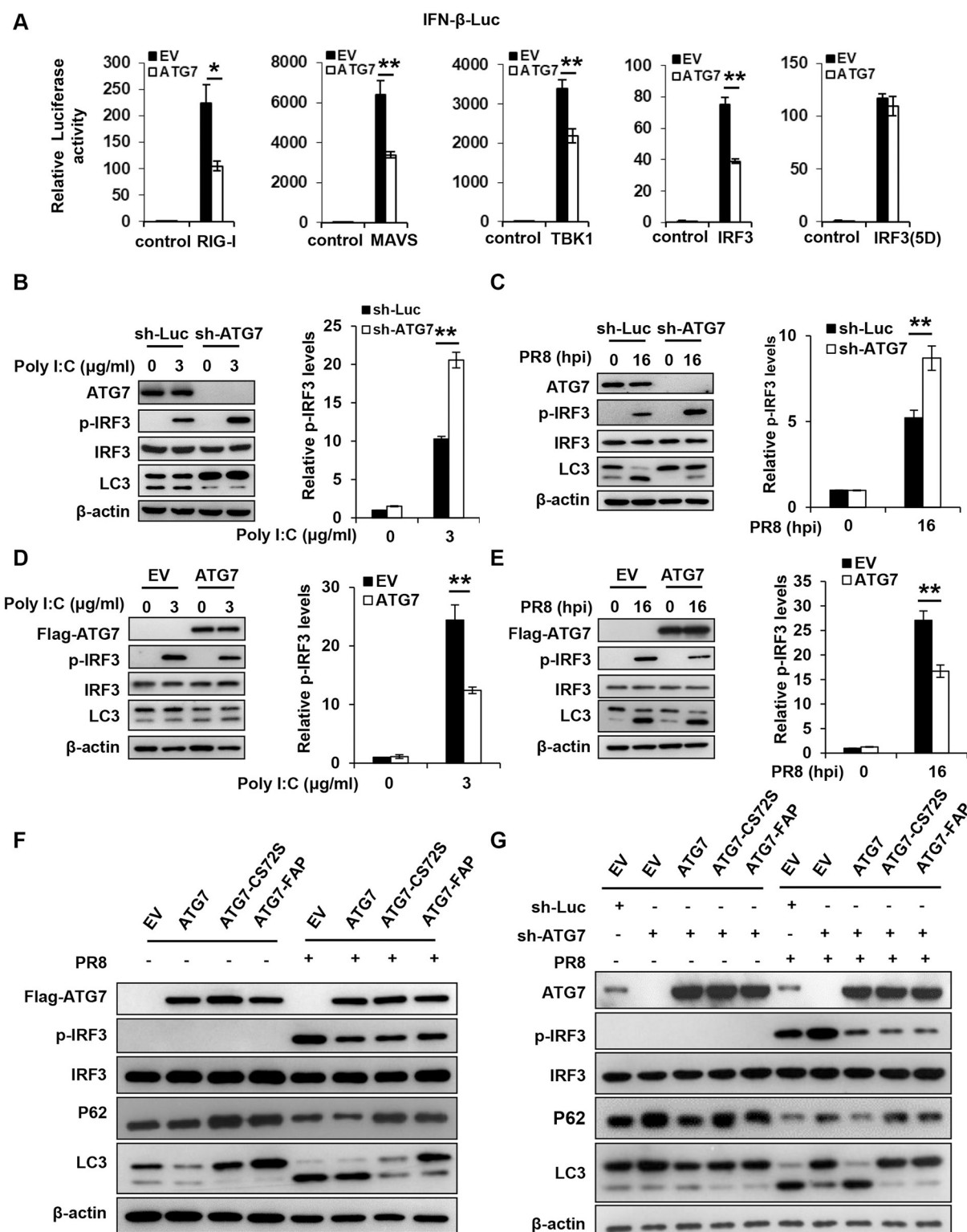

**Fig 5. ATG7 suppresses IRF3 activation in an autophagy-independent manner during the IAV infection. (A)** Control and ATG7 overexpressing 293T cells were transfected with the IFN-β luciferase reporter and either RIG-I, MAVS, TBK1, IRF3 (WT), or IRF3(5D) expressing vector. 24 hours post transfection, the cells were harvested for luciferase assay. **(B and C)** Control and ATG7 knockdown A549 cells were transfected with poly(I:C) for 4 h (B), or infected with PR8 (MOI = 0.2) for 16 h (C). The levels of IRF3 phosphorylation (p-IRF3) in the cells were examined by Western blotting. The p-IRF3 levels were quantitated by densitometry and normalized to IRF3 levels. Shown was

quantification analysis of p-IRF3 levels from three independent experiments. **(D and E)** Control and ATG7 overexpressing A549 cells were transfected with poly(I:C) for 4 h (D), or infected with PR8 (MOI = 0.2) for 16 h (E). The p-IRF3 levels in the cells were examined by Western blotting. The p-IRF3 levels were quantitated by densitometry and normalized to IRF3 levels. Shown was quantification analysis of p-IRF3 levels from three independent experiments. **(F)** Control and A549 cells overexpressing WT, or mutants of ATG7 (ATG7$^{C572S}$, ATG7$^{FAPtoDDD}$) were infected with PR8 (MOI = 0.2) for 16 h. The levels of p-IRF3 in the cells were examined by Western blotting. **(G)** Control and ATG7 knockdown A549 cells re-expressing ATG7 WT or its mutants (ATG7$^{C572S}$, ATG7$^{FAPtoDDD}$) were infected with PR8 (MOI = 0.2) for 16 h. The levels of p-IRF3 in the cells were examined by Western blotting. Western blotting data were repeated independently three times with similar results. Shown are representative data of three biologically independent experiments. Data are presented as means ± SD from three independent experiments, $^*p < 0.05$, $^{**}p < 0.01$.

cells were treated with DMSO or BAY11-7082, followed by IAV infection. Notably, the expression of GAPLINC decreased in ATG7 knockdown A549 cells compared with control cells upon IAV infection (**S6G Fig**), whereas such decrease in GAPLINC level in the ATG7 knockdown cells was impaired in the presence of BAY11-7082 (**S6G Fig**), indicating that ATG7-mediated regulation of GAPLINC is associated with NF-κB signaling in IAV-infected cells. Next, we examined the phosphorylation of p65, a subunit of NF-κB, in control and ATG7 knockdown A549 cells infected with IAV. Higher levels of phosphorylated p65 were observed in ATG7 knockdown cells than those in control cells after IAV infection (**S6H Fig**), while overexpression of ATG7 led to a significant decrease in the phosphorylation levels of p65 induced by IAV infection (**S6I Fig**). Together, these results suggest that ATG7 may positively regulate the expression of GAPLINC via inhibition of NF-κB activation.

Then, we further evaluated the function of GAPLINC during IAV infection. A549 cells stably expressing control or GAPLINC shRNAs were generated, and infected with the IAV (**Figs 6E and S6J**). It was observed that GAPLINC knockdown significantly reduced IAV titers in A549 cells, indicating that depletion of GAPLINC inhibited IAV replication (**Fig 6F**). On the other hand, we examined the effect of GAPLINC overexpression on the replication of IAV. By contrast, overexpression of GAPLINC promoted IAV replication in A549 cells, as evidenced by increased IAV titers in GAPLINC overexpressing cells (**Figs 6G, 6H and S6K**). These data reveal that altering GAPLINC expression has a significant effect on replication of IAV, and suggest that ATG7 promotes the IAV replication likely through a pathway involving GAPLINC.

Next, we further investigated the relationship between GAPLINC and ATG7 in virus-infected cells. GAPLINC was knocked down in control and ATG7 overexpressing A549 cells, followed by IAV infection (**Fig 6I**). IAV replication was measured by plaque forming assay, which showed as expected that ATG7 overexpression significantly increased IAV titers compared to the control. However, the ATG7-mediated upregulation of IAV replication was significantly attenuated by depletion of GAPLINC in host cells (**Fig 6J**). These experiments demonstrate that GAPLINC is required for the promotion of IAV replication caused by ATG7.

On the other hand, we generated ATG7 knockdown A549 cell lines overexpressing either GAPLINC or EV (**Fig 6K**). These cells were infected with IAV and examined for the viral replication. Consistently, ATG7 knockdown resulted in a significant decrease in IAV titers compared to the control, but overexpression of GAPLINC in ATG7 knockdown cells reversed the impaired IAV replication caused by loss of ATG7 (**Fig 6L**). Together, these results suggest that the lncRNA GAPLINC functions downstream of ATG7, which significantly contributes to the ATG7-mediated promotion of IAV replication.

## GAPLINC is involved in ATG7-mediated inactivation of IRF3 and inhibition of IFN response

Since lncRNA GAPLINC was identified as a critical regulator involved in the ATG7-mediated promotion of IAV replication (**Fig 6**), we evaluated the effect of GAPLNC on the activation of

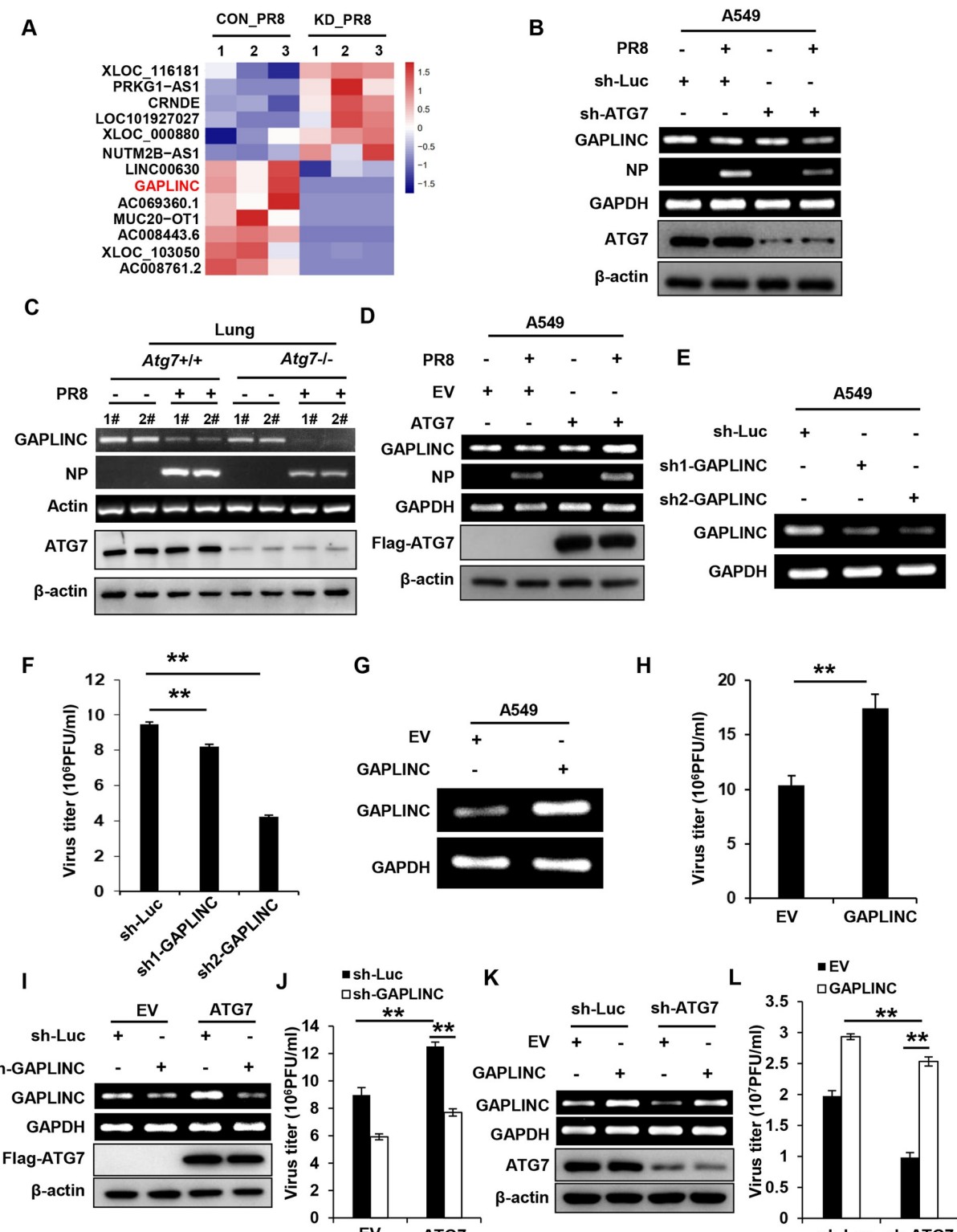

**Fig 6. Identification of lncRNA GAPLINC as a critical regulator involved in the promotion of IAV replication by ATG7. (A)**
Transcriptome RNA sequencing analysis of lncRNAs expression in control and ATG7 knockdown A549 cells infected with PR8 (MOI = 0.5) for
16 h. **(B)** RT-PCR analysis of GAPLINC RNA levels in control and ATG7 knockdown A549 cells after PR8 infection (MOI = 0.5, 16 h). **(C)** WT
and ATG7 CKO mice were infected with or without PR8 ($5\times10^4$ PFU) for 48 h. The RNA levels of GAPLINC in the lungs of mice were
examined by RT-PCR. Shown are representative data of two mice samples in each group. **(D)** GAPLINC RNA levels in ATG7 overexpressing

A549 cells after PR8 infection (MOI = 0.5, 16 h) were analyzed by RT-PCR. **(E and F)** GAPLINC RNA levels in A549 cells stably expressing control or GAPLINC shRNAs were examined by RT-PCR (E). Control and GAPLINC knockdown A549 cells were infected with PR8 (MOI = 0.2) for 16 h, and the supernatants were collected for plaque forming assay (F). **(G and H)** GAPLINC RNA levels in A549 cells infected with lentiviruses expressing EV or GAPLINC, were examined by RT-PCR (G). Control and GAPLINC overexpressing A549 cells were infected with PR8 (MOI = 0.2) for 16 h, and the supernatants were collected for plaque forming assay (H). **(I and J)** Control and ATG7 overexpressing A549 cells were infected with lentiviruses expressing control or GAPLINC shRNA. GAPLINC RNA levels and ATG7 protein levels in the cells were examined by RT-PCR and Western blotting respectively (I). The cells were infected with PR8 (MOI = 0.2) for 16 h, and the supernatants were collected for plaque forming assay (J). **(K and L)** Control and ATG7 knockdown A549 cells were infected with lentiviruses expressing EV or GAPLINC. GAPLINC RNA levels and ATG7 protein levels in the cells were examined by RT-PCR and Western blotting respectively (K). The cells were infected with PR8 (MOI = 0.2) for 16 h, and the supernatants were collected for plaque forming assay (L). RT-PCR and Western blotting data were repeated independently three times with similar results. Shown are representative data of three biologically independent experiments. Data are presented as means ± SD from three independent experiments, $^{**}p < 0.01$.

IRF3 during the IAV infection. Thus, phosphorylation levels of IRF3 were evaluated in control and GAPLINC knockdown A549 cells infected with IAV. As shown in **Fig 7A**, higher levels of phosphorylated IRF3 were observed in GAPLINC knockdown cells than in control cells after IAV infection, indicating that depletion of GAPLINC enhanced the activation of IRF3 during the viral infection. Furthermore, control and GAPLINC overexpressing A549 cells were employed to confirm this finding. Indeed, overexpression of GAPLINC caused a significant reduction of the IAV-induced phosphorylation of IRF3 (**Fig 7B**). In addition, we generated GAPLINC knockdown A549 cells re-expressing an shRNA-resistant form of GAPLINC, followed by transfection with poly(I:C) (**S7A Fig**). Consistently, higher levels of phosphorylated IRF3 were observed in GAPLINC knockdown cells than in control cells after poly(I:C) transfection, and addback of GAPLINC reversed the enhanced IRF3 phosphorylation caused by GAPLINC deficiency in the cells (**S7A Fig**). These experiments demonstrate that GAPLINC negatively regulates the IRF3 activation in IAV-infected cells. To determine the functional relevance of GAPLINC in IRF3 inactivation by ATG7 during the IAV infection, we generated ATG7 knockdown A549 cells overexpressing either GAPLINC or EV. Consistently, depletion of ATG7 resulted in a significant increase of IRF3 phosphorylation compared to the control (**Fig 7C**). Importantly, overexpression of GAPLINC in ATG7 knockdown cells reversed the enhanced IRF3 activation caused by loss of ATG7 (**Fig 7C**), indicating that depletion of ATG7 augmented the activation of IRF3 due to downregulation of GAPLINC expression in the cells. These observations suggest that lncRNA GAPLINC is involved in the negative regulation of IRF3 activation by ATG7.

The results presented above indicate that ATG7 inhibits IAV-induced expression of IFNs through inactivation of IRF3, and expression of GAPLINC contributes to the suppression of IRF3. This prompted us to address the implication of GAPLINC in regulating IFN production in host cells following IAV infection. We found that higher levels of IFN-β were detected in GAPLINC knockdown cells than those in control cells infected with IAV (**Figs 7D–7F and S7B**), while overexpression of GAPLINC caused a significant decrease in the virus-induced expression of IFN-β (**Fig 7G–7I and S7C**). These data reveal that altering GAPLINC expression has a significant impact on the IFN production, and suggest that ATG7 may restrain the IFN response through regulating GAPLINC level. To verify this, we investigated the involvement of GAPLINC in suppression of IFN response by ATG7 during the virus infection. GAPLINC was knocked down in control and ATG7 overexpressing A549 cells, followed by IAV infection (**Fig 7J**). It was observed that depletion of GAPLINC in ATG7 overexpressing cells relieved the inhibitory effect of increased ATG7 on IFN production. As shown in **Fig 7J**, ATG7 overexpression significantly decreased IFN-β levels compared with the control after IAV infection, which was reversed by depletion of GAPLINC, indicating that GAPLINC expression regulated by ATG7 contributes to the repression of IFN response by ATG7. Together, these data suggest that GAPLINC is involved in ATG7-mediated inactivation of IRF3 and inhibition of IFN response.

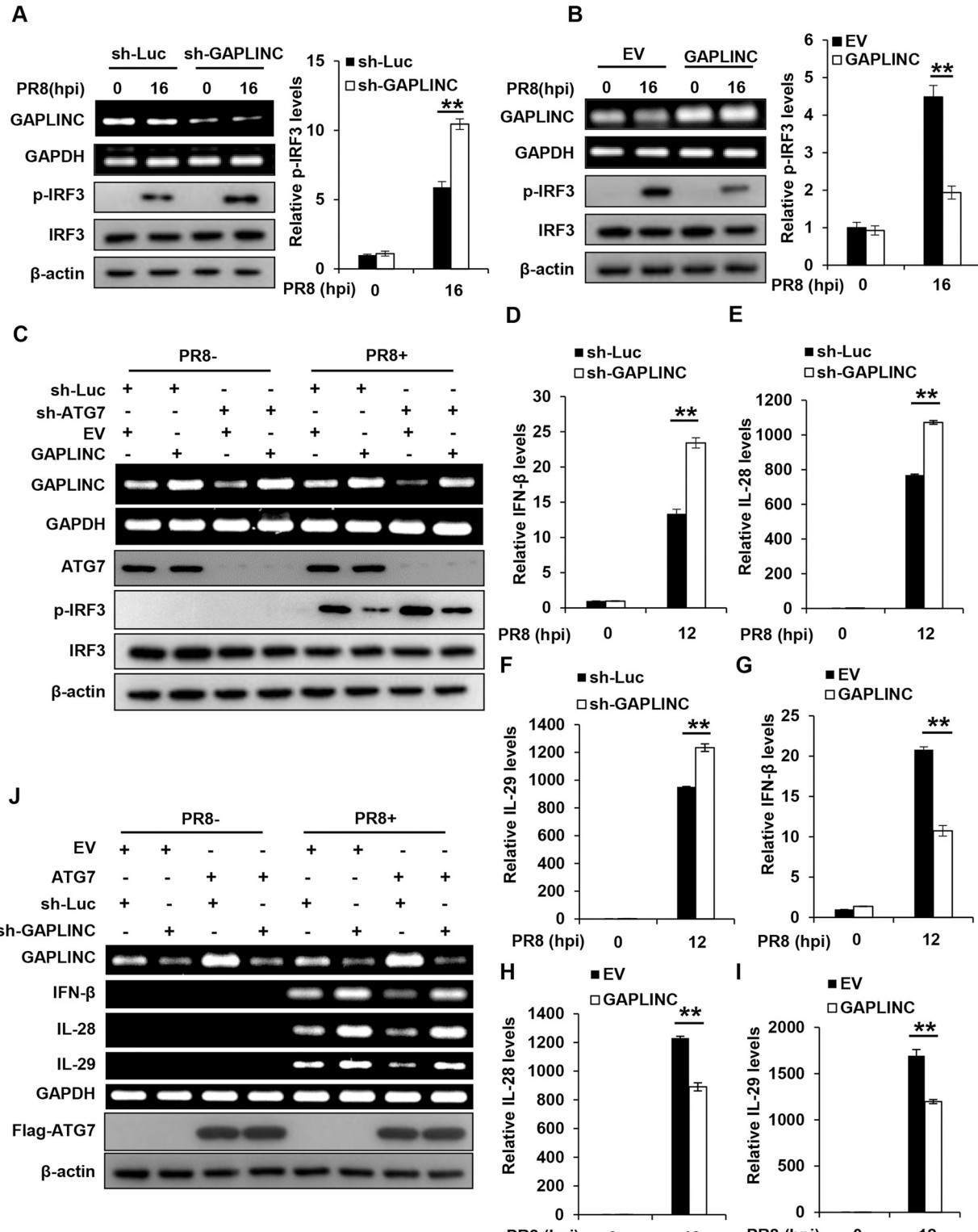

**Fig 7. GAPLINC is involved in ATG7-mediated inactivation of IRF3 and inhibition of IFN response. (A)** Control and GAPLINC knockdown A549 cells were infected with PR8 (MOI = 0.2) for 16 h. The levels of p-IRF3 in the cells were examined by Western blotting. The p-IRF3 levels were quantitated by densitometry and normalized to IRF3 levels. Shown was quantification analysis of p-IRF3 levels from three independent experiments. **(B)** Control and GAPLINC overexpressing A549 cells were infected with PR8 (MOI = 0.2) for 16 h. The p-IRF3 levels in the cells were examined by Western blotting. The p-IRF3 levels were quantitated by densitometry and normalized to IRF3 levels. Shown was

quantification analysis of p-IRF3 levels from three independent experiments. **(C)** Control and ATG7 knockdown A549 cells were infected with lentiviruses expressing EV or GAPLINC followed by infection with PR8 (MOI = 0.2) for 12 h, and p-IRF3 levels in the cells were examined by Western blotting. **(D-F)** Control and GAPLINC knockdown A549 cells were infected with PR8 (MOI = 0.5) for 12 h. The RNA levels of IFN-β (D), IL-28 (E) and IL-29 (F) were examined by qRT-PCR. **(G-I)** qRT-PCR analysis of IFN-β (G), IL-28 (H) and IL-29 (I) RNA levels in control and GAPLINC overexpressing A549 cells infected with PR8 (MOI = 0.5) for 12 h. GAPDH was chosen as a reference gene for internal standardization. **(J)** Control and ATG7 overexpressing A549 cells expressing control or GAPLINC shRNA, were infected with PR8 (MOI = 0.5) for 12 h. The IFN-β, IL-28 and IL-29 RNA levels were detected by RT-PCR. RT-PCR and Western blotting data were repeated independently three times with similar results. Shown are representative data of three biologically independent experiments. Data are presented as means ± SD from three independent experiments, $^{**}p < 0.01$.

## Discussion

Autophagy is associated with various cellular processes including cell apoptosis and differentiation. Selective autophagy involves the decoration of specific cellular components with specialized signals, thereby recruiting the autophagic machinery to the target for degradation. The involvement of autophagy in viral infections has been widely documented [7,41,42]. On the one hand, autophagy could prevent viral infection and pathogenesis by destructing viral particles, activating host innate immune responses, and coordinating adaptive immunity via facilitation of antigen presentation. On the other hand, a mass of viruses have adopted diverse strategies to evade autophagic degradation through blocking the function of host ATG proteins, or even repurpose the autophagy process for their own replication and pathogenesis. Molecular mechanisms underlying the interaction between host autophagy and IAV require further clarification.

ATG7, an essential component of autophagic machinery, which is indispensable for autophagosome formation, has been reported to play important roles in human health and diseases. For instance, clinical analysis revealed that twelve patients from five families harboring deleterious or recessive variants of ATG7 had neuro-developmental disorders [43]. The expression of ATG7 was decreased in brain tissues from amyotrophic lateral sclerosis (ALS)-frontotemporal dementia (FTD) patients, and upregulation of ATG7 could alleviate motor neuron dysfunction associated with deficiency of TARDBP/TDP-43 (TAR DNA binding protein) [44]. ATG7 has also been linked to pathogen infections but its role is highly context-dependent [45–51]. ATG7 can limit the infection of *mycobacterium tuberculosis*, human papillomavirus, Chikungunya virus, and poliovirus [45–48]. In addition, ATG7-dependent autophagy is stimulated upon infection with IAV, leading to the formation of memory CD8$^+$ T cell and endogenous presentation of epitope on MHC class II molecules [49,50]. By contrast, deficiency of ATG7 curtails hepatitis C virus (HCV) infection [51], indicating that ATG7 is required for efficient replication of HCV. Despite its importance, the function and underlying mechanism of ATG7 in IAV pathogenesis are largely unknown. In the current study, we demonstrated that ATG7 was required for efficient replication of IAV, as depletion of ATG7 decreased the viral yields *in vitro*. Furthermore, ATG7 CKO mouse model was employed to evaluate the functional relevance of ATG7 in viral pathogenesis under a more sophisticated and physiological circumstance. Loss of ATG7 rendered mice more resistant to IAV or PRV infection, as indicated by a lower degree of tissue injury, slower body weight loss, and an increased survival rate of ATG7 CKO mice after viral infection. These observations suggest an important role of ATG7 in the replication and pathogenesis of viruses such as IAV and PRV. However, further investigations are required to determine the relationship between ATG7 and PRV pathogenesis. Moreover, precise mechanisms by which these viruses manipulate ATG7 protein for their own benefits need to be better defined.

Notably, a growing number of studies have reported autophagy-independent functions of ATG proteins [24,29,52–54]. For instance, ATG16L1 promotes plasma membrane repair

through a pathway independent of autophagy, thereby preventing membrane damage by the bacterial toxin listeriolysin O and eventually restricting intercellular spread by *Listeria monocytogenes* [53]. ATG7 has been shown to regulate cell cycle arrest and prevent uncontrolled stimulation of cell death through modulating p53 activity, an autophagy-independent process during metabolic stress [55]. In the present study, we treated control and ATG7 knockdown A549 cells with autophagy inhibitors (HCQ and MRT68921) followed by IAV infection, and examined the viral replication. It was observed that ATG7 promoted IAV replication only partly requiring intact autophagy. On the other hand, the results also showed that overexpression of ATG7 attenuated the inhibitory effect of HCQ or MRT68921 on IAV replication, implying that ATG7 can promote IAV replication partly via an autophagy-independent mechanism.

In an attempt to explore the autophagy-independent mechanisms by which ATG7 regulates IAV replication, we performed RNA-Seq analysis and found that ATG7 inhibited the production of interferons (IFNs). Deficiency of ATG7 obviously enhanced the expression of type I and III IFNs in ATG7-depleted cells and mice, whereas overexpression of ATG7 impaired the interferon response to IAV infection. Furthermore, we observed that ATG7 suppressed the activation of IRF3 during IAV infection, as evidenced by significantly decreased IRF3 phosphorylation in ATG7 overexpressing cells but increased phosphorylation of IRF3 upon depletion of ATG7. Of note, re-introduction of either WT or mutants of ATG7 lacking the ability to regulate autophagy, reversed the enhanced IRF3 phosphorylation caused by loss of ATG7 in response to IAV infection. These results suggest that there exists an autophagy-independent mechanism by which ATG7 inhibits IRF3/IFN response to IAV infection. Together, the findings may explain the alleviated viral replication and pathogenesis in ATG7-depleted cells and mice.

LncRNAs have been reported to play vital roles in different biological processes including viral pathogenesis [37]. Interestingly, our RNA-Seq analysis identified that the expression of GAPLINC, an lncRNA recently characterized for its function as a negative regulator of inflammation [40], was significantly decreased in ATG7-depleted cells compared with that in control cells after IAV infection. Importantly, the enhancement of IAV replication by overexpression of ATG7 was mitigated by knockdown of GAPLINC, while overexpression of GAPLINC rescued the attenuated IAV replication caused by depletion of ATG7, implying that the lncRNA GAPLINC functions downstream of ATG7, which contributes to the ATG7-mediated promotion of IAV replication.

To gain insight into the autophagy-independent mechanism underlying action of ATG7, we further determined the functional relevance of GAPLINC in IRF3 activation and IFN response regulated by ATG7 during the IAV infection. Overexpression of GAPLINC in ATG7 knockdown cells reversed the enhanced IRF3 activation caused by loss of ATG7, while depletion of GAPLINC in ATG7 overexpressing cells relieved the inhibitory effect of increased ATG7 on IFN production, indicating lncRNA GAPLINC is critically involved in the negative regulation of IRF3 activation and IFN response by ATG7. These experiments unveil a new autophagy-independent mechanism by which ATG7 regulates viral infection via the GAPLINC-IRF3-IFN axis, which provides a understanding of how ATG7 can still influence the replication of IAV even when the cellular autophagy is blocked. To address the mechanisms whereby ATG7 and GAPLINC suppress IRF3 activation, we examined the interaction between ATG7 and IRF3, as well as GAPLINC. Interestingly, we observed that immunoprecipitation of ATG7 effectively brought down both IRF3 and GAPLINC in the cells infected with IAV (S5F and S7F Figs). This finding implies that ATG7 and GAPLINC may form a complex with IRF3 in response to IAV infection, thereby suppressing the activation of IRF3. However, further studies are needed to probe the interaction between GAPLINC and IRF3 by RNA pulldown

analysis, map specific domains responsible for the interaction of ATG7, GAPLINC, and IRF3. The precise mechanisms by which ATG7 and GAPLINC inhibit the IRF3 activation during IAV infection remain to be determined. Collectively, our studies indicate that ATG proteins have extensive biological roles beyond autophagy, and provide an important insight into the complicated interplay between host and IAV.

Further investigation is also deserved to elucidate the mechanisms underlying reciprocal regulation of autophagy-dependent and -independent pathways involved in the regulation of viral pathogenesis by ATG7. In addition, it has been reported that inflammasome activity was enhanced in ATG7 knockout mice infected with *Pseduomonas aeruginosa*, implicating a role for ATG7 in regulating inflammatory responses [56]. We found that either loss of ATG7 in mice, or depletion of GAPLINC in A549 cells, led to a significant increase in the expression of IL-6 and CXCL10 following IAV infection (S4I and S7B–S7E Figs). It is of interest to evaluate the impact of ATG7 on the inflammatory response during viral infections, and to assess the role of GAPLINC in the regulation of inflammatory responses by ATG7 in the future.

## Materials and methods

### Ethics statement

The animal experiments in this study were approved by the Research Ethics Committee of Institute of Microbiology, Chinese Academy of Sciences (Permit Number: SQIM-CAS2019033). All mouse experimental procedures were conducted in accordance with the Regulations for the Administration of Affairs Concerning Experimental Animals approved by the State Council of People's Republic of China.

### Cell lines and cell culture

A549 (human type II alveolar epithelial cells), HEK293T (human embryonic kidney cells), PK15 (porcine kidney cells), and MDCK (Madin-Darby canine kidney cells) were purchased from American Type Culture Collection (ATCC, Manassas, VA, USA). Cells were cultured in DMEM (Dulbecco's modified Eagle's medium) supplemented with 10% fetal bovine serum (FBS, Gibco, New York, NY, USA), penicillin (100 U/ml), and streptomycin (100 U/ml) at 37˚C under 5% $CO_2$ atmosphere as previously described [38].

### Generation of stable cell lines

The stable cell lines were generated using lentiviral expression systems as previously described [57]. Short hairpin RNA (shRNA)-based knockdown cell lines were generated by infection of A549 cells with lentiviruses expressing specific shRNAs in pSIH-H1-GFP vector. The sequences of shRNAs used in this study were shown in Table 1. The human GAPLINC (NCBI Accession: NR_110428.1) was cloned from A549 cells, confirmed by sequencing, and then sub-cloned into the pNL lentivirus vector. A549 cells stably expressing ATG7, ATG7$^{C572S}$ or ATG7$^{FAPtoDDD}$ were generated by infecting the cells with lentiviruses encoding indicated gene in pLVX3-FLAG vector.

### Conditional knockout of ATG7 in mice

ATG7$^{flox/flox}$/UBC-CreERT2 mice were obtained by crossing ATG7$^{flox/flox}$ mice with UBC-CreERT2 transgenic mice. The hybrid mice (ATG7$^{flox/flox}$/UBC-CreERT2) were treated with tamoxifen to generate the ATG7-deficient mice. Tamoxifen (20 mg/ml, dissolved in a mixture of 98% corn oil and 2% ethanol) was delivered into 7 weeks old ATG7$^{flox/flox}$/UBC-CreERT2 mice by intraperitoneal injection (100 μg/g body weight, once daily for 5 consecutive days) as

**Table 1. Sequences of shRNAs used in this study.**

| shRNAs | Sequences (5'-3') |
|---|---|
| sh1-ATG7 | GCTCTTCCTTACTTCTTAATC |
| sh2-ATG7 | GCAAATGAGATATGGGAATCC |
| sh1-GAPLINC | GGTTTCCTAAGTCTTATGACT |
| sh2-GAPLINC | GCCAATGCCTGAAATAATGAA |
| sh-Luciferase | CTTACGCTGAGTACTTCGA |

previously described [58]. All mice were housed and bred in the animal facility at the Institute of Microbiology, Chinese Academy of Sciences, under specific pathogen free conditions. ATG7$^{flox/flox}$ mice were purchased from the Nanjing Biomedical Research Institute of Nanjing University (NBRI, Nanjing, China).

## Viruses, virus infection and virus titration

Influenza virus A/WSN/1933 (H1N1), A/Puerto Rico/8/1934 (H1N1), and Sendai virus (SeV) were propagated in specific pathogen free (SPF) embryonated chicken eggs as previously described [59]. Herpes simplex virus type 1 (HSV-1) was propagated in Vero cells [38]. Pseudorabies virus (PRV) was propagated in MDCK cells [60]. WSN, PR8, H9N2, SeV, or HSV-1 was used to infect A549 cells. PRV was used to infect PK15 cells. Cells were incubated with the virus for 1 h at 37°C and cultured in DMEM containing 2 g/ml trypsin for the indicated time. Tamoxifen was delivered into 7 weeks old ATG7$^{flox/flox}$/UBC-CreERT2 mice by intraperitoneal injection as described above. One day after the last tamoxifen treatment, mice were inoculated intranasally with PR8, or were injected intramuscularly with PRV as previously described [58]. Virus replication was titrated by hemagglutinin and plaque forming assays [61]. Briefly, MDCK cells infected with serial dilutions of the supernatants of cell cultures were washed with PBS and overlaid with α-minimal essential medium containing low-melting-point agarose (Promega, Madison, WI, USA) and TPCK (tolylsulfonyl phenylalanyl chloromethyl ketone)-treated trypsin (Sigma-Aldrich, St. Louis, MO, USA). After incubation for 72 h, plaques were stained and counted. Lung virus titers were determined 72 h post-infection. Lungs of infected mice were homogenized in 1 mL of ice-cold PBS and frozen at 80°C for 14 h. Then, thawed samples were centrifuged at 2,000 × $g$ for 10 min, and the supernatants were titrated by plaque assay as described above.

## RNA extraction, RT-PCR and qRT-PCR

Total RNA was isolated from cells or tissues using TRIzol reagent (Invitrogen, Carlsbad, CA, USA). cDNA was synthesized using GoScript reverse transcriptase (Promega, Madison, WI, USA), followed by PCR using rTaq DNA polymerase, or quantitative PCR using KAPA HRM FAST qPCR Master Mix (2X) Kits (KAPA BIOSYSTEMS, Indianapolis, IN, USA) with specific primers. The sequences of primers used in this study were shown in Table 2. For quantification, the 2$^{-\Delta\Delta Ct}$ method was used to calculate relative RNA levels against GAPDH or actin [62].

## Western blotting, immunoprecipitation and RNA immunoprecipitation

Western blotting was performed as described previously [63]. Briefly, cell or tissue samples were separated on SDS-polyacrylamide gel, transferred onto a nitrocellulose membrane, and probed with antibodies as indicated. Immunoprecipitation (IP) was performed as previously described [59]. RNA immunoprecipitation (RIP) was performed using Magna RIP RNA-Binding Protein Immunoprecipitation Kit (Millipore, Burlington, MA, USA) according to the manufacturer's

**Table 2. Sequences of primers used in this study.**

| Gene Name | Sequences |
|---|---|
| *ATG7* (human) | Forward: AGTGTCACTCTGGAGCAAGC<br>Reverse: AAAAAGCGATGAGCCCAGGA |
| *ATG7* (mouse) | Forward: TCGAAAACCCCATGCTCCTC<br>Reverse: AGGGCCTGGATCTGTTTTGG |
| *GAPLINC* (human) | Forward: TGACACATCCTCTTGGTTTCCT<br>Reverse: TCTGTGCATACCCTGAGTCC |
| *GAPLINC*<br>(mouse) | Forward: ACCTTGTAAGCTTGGAGAACCC<br>Reverse: GTGCCGAAATTCCAAAGGCA |
| *IFN-β* (human) | Forward: GCTCTCCTGTTGTGCTTCTCCAC<br>Reverse: CAATAGTCTCATTCCAGCCAGTGC |
| *IFN-β* (mouse) | Forward: GGTCCGAGCAGAGATCTTCA<br>Reverse: CACTACCAGTCCCAGAGTCC |
| *IL-28* (human) | Forward: CCACATAGCCCAGTTCAA<br>Reverse: AAGCGACTCTTCTAAGGCA |
| *IL-28* (mouse) | Forward: AGCTGCAGGCCTTCAAAAAG<br>Reverse: TGGGAGTGAATGTGGCTCAG |
| *IL-29* (human) | Forward: TGGTGACTTTGGTGCTAGGC<br>Reverse: GGCCTTCTTGAAGCTCGCTA |
| *Sev-NP* | Forward: ATAAGTCGGGAGGAGGTGCT<br>Reverse: GTTGACCCTGGAAGAGTGGG |
| *IAV-NP* | Forward: TCAAACGTGGGATCAATG<br>Reverse: GTGCAGACCGTGCTAGAA |
| *PRV-gE* | Forward: CTTCCACTCGCAGCTCTTCT<br>Reverse: TAGATGCAGGGCTCGTACAC |
| *Actin* (mouse) | Forward: GCTGCCTCAACACCTCAACCC<br>Reverse: GTCCCTCACCCTCCCAAAAG |
| *GAPDH* (human) | Forward: AGAAGGCTGGGGCTCATTTG<br>Reverse: AGGGGCCATCCACAGTCTTC |

instructions. The following antibodies were used in this study: ATG7 (Proteintech, Wuhan, China), IRF3 (Cell Signaling Technology, Danvers, MA, USA), phospho-IRF3 (S386) (abcam, Cambridge, UK), P65 (Cell Signaling Technology, Danvers, MA, USA), phospho-P65 (S536) (Cell Signaling Technology, Danvers, MA, USA), and actin (Santa Cruz Biotechnology, Dallas, TX, USA). The antibody against IAV NP was generated and used as previously described [58]. The following reagents were used in this study: dimethyl sulfoxide (DMSO) (Sigma-Aldrich), hydroxychloroquine Sulfate (HCQ) (Selleck, Houston, TX, USA), and MRT68921 (Selleck).

## Dual-luciferase assay

The 293T cells were co-transfected with IFN-β luciferase reporter, pRL-TK and the plasmid encoding RIG-I, MAVS, TBK1, IRF3 or IRF3 (5D). At 24 h post-transfection, luciferase activity was measured using the dual-luciferase reporter assay system (Promega, USA) and a Luminoskan Ascent luminometer (Thermo Scientific, USA) according to the manufacturer's instructions. The relative luciferase expression was determined by the normalization of the firefly luciferase activity to that of Renilla luciferase. The luciferase reporter plasmids used in this study were provided by Dr. Chunfu Zheng (Fujian Medical University, Fuzhou, China) [64].

## RNA-Seq analysis

Control and ATG7 knockdown A549 cells were infected with PR8 for 16 h. Total RNAs were isolated from three independent groups of the cells, and RNA integrity was evaluated using the

Agilent 2100 Bioanalyzer (Agilent Technologies, Santa Clara, CA, USA). The libraries were constructed using TruSeq Stranded Total RNA with Ribo-Zero Gold, and then sequenced on the Illumina sequencing platform. At the data procession step, Trimmomatic software was used for adapter removing. Sequenced reads were masked for low-quality sequence, and then clean reads were mapped to hg38 whole genome using HISAT2. The FPKM (Fragments Per Kilobase Million, Fragments Per Kilobase of exon model per Million mapped fragments) value and counts value were obtained using eXpress. Differential expression was analyzed using DESeq. The RNA-Seq data have been deposited in GEO public database under the accession number GSE211357.

### Histopathological analysis

Mouse tissues were fixed in 4% paraformaldehyde and embedded in paraffin. Then, 4 mm-thick sections were prepared and stained with hematoxylin and eosin (HE). The slides were visualized under an Olympus BH-2 microscope (Tokyo, Japan). Lung tissue damage caused by IAV infection were evaluated and scored according to inflammatory cell infiltration, alveolar edema, alveolar hemorrhage and thickening of alveolar wall [57].

### Statistical analysis and reproducibility

Statistical significance was determined by Student's t-test. All data represent the mean ± SD of at least three independent experiments, and $p$ value $< 0.05$ was considered to be statistically significant. Western blotting and RT-PCR assays shown here were successfully repeated at least three times.

### Supporting information

**S1 Fig. Altering ATG7 expression has profound effects on the replication of IAV and SeV. (A)** ATG7 protein levels in A549 cells stably expressing control or ATG7 shRNAs were examined by Western blotting. **(B and C)** Control and ATG7 knockdown A549 cells were infected with SeV (MOI = 0.5) for the indicated time. Viral NP RNA levels in the cells were examined by RT-PCR (B) and qRT-PCR (C) respectively. **(D and E)** Control and ATG7 overexpressing A549 cells were infected with WSN (MOI = 0.5) (D), or PR8 (MOI = 0.2) for 16 h (E). The supernatants were collected for plaque forming assay. **(F and G)** Control and ATG7 overexpressing A549 cells were infected with SeV (MOI = 0.5) for the indicated time. RT-PCR (F) and qRT-PCR (G) were performed to test viral NP RNA levels in the cells. Data are presented as means ± SD from three independent experiments, $^{**}p < 0.01$.
(PDF)

**S2 Fig. Conditional knockout of ATG7 significantly reduces virulence of PRV in mice. (A-D)** WT and ATG7 CKO mice were injected intramuscularly with $1 \times 10^6$ PFU of PRV. Viral gE RNA levels in the brains (A) and lungs (B) from WT and ATG7 CKO mice at 2 dpi were tested by qRT-PCR. Viral gE protein levels in the livers from WT and ATG7 CKO mice at 2 dpi were determined by Western blotting (C), and gE RNA levels were tested by RT-PCR (C) and qRT-PCR (D) respectively. RT-PCR and Western blotting data were repeated independently three times with similar results. Shown are representative data of three biologically independent experiments. Data are presented as means ± SD from three independent experiments, $^{*}p < 0.05, ^{**}p < 0.01$.
(PDF)

**S3 Fig. ATG7 promotes the replication of IAV in autophagy-dependent and -independent manners. (A and B)** Control and ATG7 knockdown A549 cells were pretreated with HCQ

(20 μM) (A) or MRT68921 (5 μM) (B) for 3 h, and then infected with PR8 (MOI = 0.2) for 16 h. The levels of p62 and LC3 in the cells were detected by Western blotting. **(C and D)** Control and ATG7 knockdown A549 cells were transfected with siRNA targeting BECN1 (C) or ATG3 (D), followed by infection with PR8 (MOI = 0.2) for 16 h. The RNA levels of viral NP were detected by RT-PCR. **(E and F)** Control and ATG7 overexpressing A549 cells were transfected with siRNA targeting BECN1 (E) or ATG3 (F), followed by infection with PR8 (MOI = 0.2) for 16 h. The RNA levels of viral NP were detected by RT-PCR. RT-PCR and Western blotting data were repeated independently three times with similar results. Shown are representative data of three biologically independent experiments.
(PDF)

**S4 Fig. Altering ATG7 expression has profound effects on the production of IFNs. (A)** Transcriptome RNA sequencing analysis of control and ATG7 knockdown A549 cells infected with or without PR8 (MOI = 0.5) for 16 h. **(B and C)** Control and ATG7 knockdown A549 cells were infected with PR8 (MOI = 0.2) (B), or WSN (MOI = 0.5) (C) for the indicated time. The RNA levels of IFN-β, IL-28 and IL-29 were examined by RT-PCR. **(D-G)** Control and ATG7 knockdown A549 cells were infected with SeV (MOI = 0.5) for the indicated time. The RNA levels of IFN-β, IL-28 and IL-29 were examined by RT-PCR (D) and qRT-PCR (E-G) respectively. **(H)** Control and ATG7 overexpressing A549 cells were infected with WSN (MOI = 0.5) for the indicated time. The RNA levels of IFN-β, IL-28 and IL-29 were examined by RT-PCR. **(I)** WT and ATG7 CKO mice were infected with PR8 ($5\times10^4$ PFU) for 0, 24, and 48 h. The RNA levels of IFN-β, IL-28, IL-6 and CXCL10 in the lungs of mice were examined by RT-PCR. RT-PCR and Western blotting data were repeated independently three times with similar results. Shown are representative data of three biologically independent experiments. Data are presented as means ± SD from three independent experiments, $^{**}p < 0.01$.
(PDF)

**S5 Fig. ATG7 suppresses IRF3 activation in an autophagy-independent manner during the IAV infection. (A)** Control and ATG7 knockdown A549 cells were infected with SeV for 12 h, and then subjected to immunofluorescence assays with anti-IRF3 antibody (left). Nuclei were visualized with DAPI (blue). Shown are percentages of cells with IRF3 located in the nuclei (right). Data are presented as means ± SD, $^{**}p < 0.01$. **(B)** A549 cells were transfected with control siRNA or BECN1 siRNA, followed by infection with PR8 (MOI = 0.2) for 16 h. The phosphorylated IRF3 (p-IRF3) levels in the cells were examined by Western blotting. **(C)** A549 cells were transfected with control siRNA or ATG3 siRNA, followed by infection with PR8 (MOI = 0.2) for 16 h, and the p-IRF3 levels in the cells were examined by Western blotting. **(D-E)** Control and ATG7 knockdown A549 cells re-expressing an shRNA-resistant form of ATG7, were transfected with poly(I:C) for 4 h. The levels of p-IRF3 in the cells were examined by Western blotting. **(F)** Control and ATG7 overexpressing A549 cells were infected with PR8 (MOI = 0.5) for 12 h, followed by immunoprecipitation (IP) assays using Flag antibody. IRF3 and Flag-ATG7 levels were examined by Western blotting.
(PDF)

**S6 Fig. ATG7 enhances the expression of GAPLINC via inhibition of NF-κB activation. (A)** Transcriptome RNA sequencing analysis of lncRNAs expression in control and ATG7 knockdown A549 cells infected with or without PR8 (MOI = 0.5) for 16 h. **(B and C)** A549 cells were transfected with BECN1 siRNA (B) or ATG3 siRNA (C), followed by infection with PR8 (MOI = 0.2) for 16 h. GAPLINC RNA levels in the cells were examined by RT-PCR. **(D and E)** A549 cells were infected with PR8 at indicated MOIs for 16 h (D), or at an MOI of 0.2 for the indicated time (E). RT-PCR and Western blotting were performed to examine

GAPLINC RNA levels and ATG7 protein levels respectively. **(F)** A549 cells were pretreated with BAY-117082 (50 nmol/ml) or DMSO for 1 h, followed by infection with PR8 (MOI = 0.2) for the indicated time. The RNA levels of GAPLINC were examined by RT-PCR. **(G)** Control and ATG7 knockdown A549 cells were pretreated with BAY-117082 (50 nmol/ml) or DMSO for 1 h, followed by infection with PR8 (MOI = 0.2) for 16 h. The GAPLINC RNA levels were detected by RT-PCR. **(H)** Control and ATG7 knockdown A549 cells were infected with PR8 (MOI = 0.2) for 16 h. The levels of P65 phosphorylation (p-P65) in the cells were examined by Western blotting. **(I)** Control and ATG7 overexpressing A549 cells were infected with PR8 (MOI = 0.2) for 16 h. The levels of p-P65 in the cells were examined by Western blotting. **(J)** GAPLINC RNA levels in A549 cells stably expressing control or GAPLINC shRNAs were examined qRT-PCR. **(K)** GAPLINC RNA levels in A549 cells infected with lentiviruses expressing EV or GAPLINC, were examined by qRT-PCR. Data are presented as means ± SD from three independent experiments, $^{**}p < 0.01$.
(PDF)

**S7 Fig. Altering GAPLINC expression has profound effects on the production of IFNs, IL-6 and CXCL10. (A)** Control and GAPLINC knockdown A549 cells re-expressing an shRNA-resistant form of GAPLINC were transfected with poly(I:C) for 4 h. The levels of p-IRF3 in the cells were examined by Western blotting. **(B)** RT-PCR analysis of IFN-β, IL-28, IL-29, IL-6 and CXCL10 RNA levels in control and GAPLINC knockdown A549 cells infected with PR8 (MOI = 0.5) for 12 h. **(C)** The mRNA levels of IFN-β, IL-28, IL-29, IL-6 and CXCL10 in control and GAPLINC overexpressing A549 cells infected with PR8 (MOI = 0.5) for 12 h, were detected by RT-PCR. **(D)** Control and GAPLINC knockdown A549 cells were infected with PR8 (MOI = 0.5) for 12 h. The RNA levels of IL-6 and CXCL10 were examined by qRT-PCR. **(E)** qRT-PCR analysis of IL-6 and CXCL10 RNA levels in control and GAPLINC overexpressing A549 cells infected with PR8 (MOI = 0.5) for 12 h. **(F)** Control and ATG7 overexpressing A549 cells were infected with PR8 (MOI = 0.5) for 16 h, and then subjected to RNA immunoprecipitation (RIP) assays using Flag antibody. GAPDH served as the negative control. RT-PCR data were repeated independently three times with similar results. Shown are representative data of three biologically independent experiments. Data are presented as means ± SD from three independent experiments, $^{*}p < 0.05$, $^{**}p < 0.01$.
(PDF)

## Acknowledgments

We thank all the members of Chen laboratory for helpful discussions and assistance.

## Author Contributions

**Conceptualization:** Biao Chen, Guijie Guo, Ji-Long Chen.

**Data curation:** Biao Chen, Guijie Guo, Guoqing Wang, Qianwen Zhu, Lulu Wang, Wenhao Shi, Song Wang, Yuhai Chen, Xiaojuan Chi, Faxin Wen, Mohamed Maarouf, Zhou Yang, Ji-Long Chen.

**Formal analysis:** Biao Chen, Guijie Guo, Guoqing Wang, Ji-Long Chen.

**Funding acquisition:** Ji-Long Chen.

**Investigation:** Biao Chen, Guijie Guo, Guoqing Wang, Qianwen Zhu, Lulu Wang, Wenhao Shi, Yuhai Chen, Xiaojuan Chi, Faxin Wen, Mohamed Maarouf, Ji-Long Chen.

**Methodology:** Biao Chen, Guijie Guo, Guoqing Wang, Ji-Long Chen.

**Project administration:** Biao Chen, Guijie Guo, Ji-Long Chen.

**Resources:** Ji-Long Chen.

**Software:** Biao Chen, Guoqing Wang.

**Supervision:** Biao Chen, Guijie Guo, Ji-Long Chen.

**Validation:** Biao Chen, Guijie Guo, Guoqing Wang, Qianwen Zhu, Lulu Wang, Wenhao Shi, Song Wang, Yuhai Chen, Zhou Yang, Ji-Long Chen.

**Visualization:** Biao Chen, Guijie Guo, Guoqing Wang, Qianwen Zhu, Lulu Wang, Wenhao Shi, Song Wang, Xiaojuan Chi, Faxin Wen, Mohamed Maarouf, Zhou Yang, Ji-Long Chen.

**Writing – original draft:** Biao Chen, Guijie Guo, Ji-Long Chen.

**Writing – review & editing:** Biao Chen, Guijie Guo, Shile Huang, Ji-Long Chen.

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
