## [Decision Letter · Decision Letter 0]

14 Oct 2023

Dear Professor Chen,

Thank you very much for submitting your manuscript "ATG7/GAPLINC/IRF3 axis plays a critical role in regulating pathogenesis of influenza A virus" for consideration at PLOS Pathogens. As with all papers reviewed by the journal, your manuscript was reviewed by members of the editorial board and by several independent reviewers. In light of the reviews (below this email), we would like to invite the resubmission of a significantly-revised version that takes into account the reviewers' comments.

Please note that new experiments, new results and extensive text revisions are required to address reviewer comments. This includes experiments addressing mechanisms by which ATG7-GAPLINC reduce IRF3 phosphorylation, tests for effects of ATG genes beyond ATG7 in controlling IAV, and additional control tests involving CRISPR KO and complementation assays that further determine credibility of findings. Text needs to be significantly revised in ways that reveal clear premises and logical progressions of the work as it builds a case for the ATG7-GAPLINC-IRF3 axis in IAV infection.

We cannot make any decision about publication until we have seen the revised manuscript and your response to the reviewers' comments. Your revised manuscript is also likely to be sent to reviewers for further evaluation.

Sincerely,

Tom Gallagher

Guest Editor

PLOS Pathogens

Matthias Schnell

Section Editor

PLOS Pathogens

Kasturi Haldar

Editor-in-Chief

PLOS Pathogens

orcid.org/0000-0001-5065-158X

Michael Malim

Editor-in-Chief

PLOS Pathogens

orcid.org/0000-0002-7699-2064

Please note that new experiments, new results and extensive text revisions are required to address reviewer comments. This includes experiments addressing mechanisms by which ATG7-GAPLINC reduce IRF3 phosphorylation, tests for effects of ATG genes beyond ATG7 in controlling IAV, and additional control tests involving CRISPR KO and complementation assays that further determine credibility of findings. Text needs to be significantly revised in ways that reveal clear premises and logical progressions of the work as it builds a case for the ATG7-GAPLINC-IRF3 axis in IAV infection.

Reviewer's Responses to Questions

**Part I - Summary**

Reviewer #1: In this study by Chen et al., the authors investigate the role of autophagy protein ATG7 on influenza A virus (IAV) replication and cell autonomous immune responses. The authors report that shRNA knockdown of ATG7 in the human lung epithelial cell line A549 decreased IAV replication. In contrast, overexpression (OE) of ATG7 in A549 cells increased IAV replication, indicating that ATG7 promotes IAV replication. These findings are further supported by in vivo studies in a tamoxifen inducible Atg7 ko mouse model, where loss of Atg7 decreased IAV viral titer in the lungs. Through RNASeq analysis, the authors identified a non-coding RNA GAPLINC that was lower in IAV infected ATG7 KD cells. shRNA KD or OE of GAPLINC decreased or increased IAV replication, respectively. Interestingly, OE of GAPLINC in ATG7 KD cells was sufficient to rescue IAV replication, indicating that GAPLINC acts downstream of ATG7 and that the lowered IAV replication in ATG7 KD cells is due to decreased GAPLINC levels. Next, the authors investigated if ATG7 and GAPLINC promotes replication by reducing RLR signaling and report that ATG7 suppresses the phosphorylation of IRF3, thereby promoting IAV replication. Similarly, the non-coding RNA GAPLINC reduced the levels of p-IRF3. The authors conclude that ATG7 and GAPLINC promote IAV replication by suppressing IRF3 activation. Although these are interesting findings, the study lacks mechanistic insights into how GAPLINC suppresses IRF3 activation. In addition, the study relies on observations with ATG7 KD/ Atg7 KOs, which needs to be supported by KD or KO of other autophagy genes. Furthermore, some of the RLR suppression studies need to be validated with CRISPR KO or complemented KD cells with shRNA resistant cDNA of ATG7 and GAPLINC.

Reviewer #2: Biao Chen et., al demonstrate ATG7, an essential autophagy effector enzyme, suppresses IRF3 activation and interferon production via lncRNA GAPLINC, revealing an autophagy-independent mechanism whereby ATG7 restrains host innate immunity and unveiling a critical role of ATG7/GAPLINC/IRF3 axis in regulating IAV pathogenesis. These results show that ATG7 has multiple biological roles beyond autophagy, and provide an important insight into the complicated interplay between host and IAV. In general, there are many points involved in the manuscript with large workload and the experimental results are clear and the evidence is relatively sufficient.

**Part II – Major Issues: Key Experiments Required for Acceptance**

Reviewer #1: Major concerns:

1. The lack of mechanistic insights into how ATG7 and GAPLINC suppress IRF3 activation. The role of autophagy in RLR signaling and GAPLINC in the suppression of inflammatory gene expression has been reported by others. I think further investigation of the underlying mechanisms will strengthen the manuscript.

2. It is unclear how the expression of GAPLINC is lowered by loss of ATG7.

3. It needs to be demonstrated if there are changes in ATG7 and GAPLINC during IAV infection and how this impacts the overall IAV life cycle.

4. Are the levels of Gaplinc lower in Atg7 KO mouse lungs? The findings will be strengthened if the authors could demonstrate their findings from A549 cells in mouse Atg7 ko cells.

5. The data for autophagy independent role of ATG7 is weak. The majority of the data relies on chemical inhibitors, which can also block IAV infection by blocking endosomal acidification (HCQ).

Reviewer #2: Major comments:

The logic of the article is a bit confusing and needs to be revised. For example, the authors found that ATG7 could promote IAV infection in an autophagy-independent manner, and then found and explained that lncRNA GAPLINC was downregulated in ATG7 knockdown A549 cells immediately. Later, the authors directly jumped to the context of innate immunity response by ATG7 and guess lncRNA GAPLINC may play critical roles in it. Finally, the relationship between these three was linked together by all the authors’ suppose instead of experimental results. As a result, the research content does not demonstrate a hierarchical and closely connected logic.

Actually, when the authors found that ATG7 could promote IAV infection in an autophagy-independent manner and in order to clarify this molecular mechanism, RNA-Seq experiments were conducted. Compared with RNA-seq results of ATG7 wild-type and knockdown cells, finding that interferon type I and III were upregulated in ATG7 knockdown cells, indicating that ATG7 may regulate IAV infection by influencing innate immune response. Then, experiments were conducted to demonstrate that ATG7 could assist in IAV infection by inhibiting the phosphorylation of IRF3. Next, how does ATG7 regulate this process? Then according to RNA-seq results and it came to the relevant content of lncRNA GAPLINC. This kind of logic is more compact and rigorous.

**Part III – Minor Issues: Editorial and Data Presentation Modifications**

Reviewer #1: Minor concerns:

1. Most of the data presented in the manuscript does not indicate the time points for analysis (hpi or dpi?)

2. For hemagglutination titers, the y-axis label should say HA titer and not viral titers.

3. There is a two-fold difference in viral titers. Surprisingly, the NP levels on the western blot are so different.

4. Nomenclature for mouse gene names are incorrect.

5. It will be informative for the readers to show the individual mouse lung titers as a scatter plot instead of a bar graph.

6. Timeline between Tamoxifen injection and viral infection is not clear.

7. Some of the graphs are overlapping and the legends/numbering are blocked.

8. GAPLINC- RNASeq data doesn’t include mock controls. It is important to include this data as prior studies indicate that KD of GAPLINC increases the basal expression levels of inflammatory cytokines.

Reviewer #2: Minor comments:

1. S1A Fig showed that A549 cells expressing ATG7 shRNAs could still detect ATG7 protein, but in Fig 3G or S3B Fig, A549 cells expressing ATG7 shRNAs seemed appeared no ATG7, just like ATG7 knockout cells. It needs to be modified and explained.

2. In Fig 2A, authors detected ATG7 protein levels in ATG7 flox/flox/UBC-CreERT2 mice tissues, but instead of thymus, all the tissues showed a high level of ATG7 in ATG7-/- mice. I don't know whether “ ATG7 knockout mice” described in the article are suitable.

3. In the context of effects of ATG7 on the IRF3 activation, authors only checked IRF3 phosphorylation. What’s the effects of ATG7 on IRF3 dimerization and nuclear translocation. Authors should add these results to make the article conclusion stronger.

PLOS authors have the option to publish the peer review history of their article (what does this mean?). If published, this will include your full peer review and any attached files.

Reviewer #1: No

Reviewer #2: No
---

## [Decision Letter · Decision Letter 1]

8 Jan 2024

Dear Professor Chen,

We are pleased to inform you that your manuscript 'ATG7/GAPLINC/IRF3 axis plays a critical role in regulating pathogenesis of influenza A virus' has been provisionally accepted for publication in PLOS Pathogens.

Best regards,

Tom Gallagher

Guest Editor

PLOS Pathogens

Matthias Schnell

Section Editor

PLOS Pathogens

Kasturi Haldar

Editor-in-Chief

PLOS Pathogens

orcid.org/0000-0001-5065-158X

Michael Malim

Editor-in-Chief

PLOS Pathogens

orcid.org/0000-0002-7699-2064

Reviewer Comments (if any, and for reference):

Reviewer's Responses to Questions

**Part I - Summary**

Reviewer #1: The authors have addressed all of my previous concerns.

**Part II – Major Issues: Key Experiments Required for Acceptance**

Reviewer #1: (No Response)

**Part III – Minor Issues: Editorial and Data Presentation Modifications**

Reviewer #1: (No Response)

PLOS authors have the option to publish the peer review history of their article (what does this mean?). If published, this will include your full peer review and any attached files.

Reviewer #1: No

---

## [Editor Report · Acceptance letter]

12 Jan 2024

Dear Professor Chen,

We are delighted to inform you that your manuscript, "ATG7/GAPLINC/IRF3 axis plays a critical role in regulating pathogenesis of influenza A virus," has been formally accepted for publication in PLOS Pathogens.

Best regards,

Michael Malim

Editor-in-Chief

PLOS Pathogens

orcid.org/0000-0002-7699-2064